# Biomimetic electromechanical stimulation to maintain adult myocardial slices in vitro

Samuel A. Watson[1], James Duff[1], Ifigeneia Bardi[1], Magdalena Zabielska[2], Santosh S. Atanur[3], Richard J. Jabbour[1], André Simon[4], Alejandra Tomas[5], Ryszard T. Smolenski[2], Sian E. Harding [1], Filippo Perbellini [1] & Cesare M. Terracciano[1]

Adult cardiac tissue undergoes a rapid process of dedifferentiation when cultured outside the body. The in vivo environment, particularly constant electromechanical stimulation, is fundamental to the regulation of cardiac structure and function. We investigated the role of electromechanical stimulation in preventing culture-induced dedifferentiation of adult cardiac tissue using rat, rabbit and human heart failure myocardial slices. Here we report that the application of a preload equivalent to sarcomere length $(SL) = 2.2\,\mu m$ is optimal for the maintenance of rat myocardial slice structural, functional and transcriptional properties at 24 h. Gene sets associated with the preservation of structure and function are activated, while gene sets involved in dedifferentiation are suppressed. The maximum contractility of human heart failure myocardial slices at 24 h is also optimally maintained at $SL = 2.2\,\mu m$. Rabbit myocardial slices cultured at $SL = 2.2\,\mu m$ remain stable for 5 days. This approach substantially prolongs the culture of adult cardiac tissue in vitro.

---

[1] National Heart & Lung Institute, Imperial College London, London W12 0NN, UK. [2] Department of Biochemistry, Medical University of Gdańsk, Gdańsk 80-210, Poland. [3] Faculty of Medicine, Department of Medicine, Imperial College London, London SW7 2AZ, UK. [4] Department of Cardiothoracic Transplantation & Mechanical Circulatory Support, Royal Brompton & Harefield NHS Foundation Trust, Harefield UB9 6JH, UK. [5] Section of Cell Biology and Functional Genomics, Division of Diabetes, Endocrinology and Metabolism, Department of Medicine, Imperial College London, London W12 0NN, UK. Correspondence and requests for materials should be addressed to F.P. (email: f.perbellini@imperial.ac.uk) or to C.M.T. (email: c.terracciano@imperial.ac.uk)

Conventional in vitro cell culture systems are relatively accessible and have been indispensable in clarifying and corroborating mechanisms, but their oversimplified nature only distantly recapitulates the in vivo environment[1,2]. An in vitro model of intermediate complexity, with preserved cardiac multicellularity, architecture and physiology that can be maintained in a stable manner in culture and bridges the gap between cellular and in vivo studies is greatly needed.

Myocardial slices are highly viable, ultrathin (300-μm) slices of the ventricular myocardium that retain the native multicellularity, architecture and physiology of the heart[3]. They are a platform of intermediate complexity, allowing the modulation of the myocardium at the macroscopic level (biological compounds, drugs and mechanical load), while studying cellular responses within a native cardiac environment. Our group has recently optimised the technique, resulting in a preparation with 97% viable cardiomyocytes, robust contractile function and physiological $Ca^{2+}$ handling and conduction velocities[3]. It is also possible to produce many slices from the same heart, thus reducing the number of animals/biopsies required in studies, and slices can be produced from human biopsies providing a clinically relevant human model in vitro system[4-6]. The thinness of myocardial slices ensures the diffusion of oxygen and other small molecules to all cells, which is very difficult to achieve with other preparations and is crucial for chronic culture[7,8]. However, our group and others have demonstrated that current culture techniques, utilising a liquid–air interface, cannot prevent tissue dedifferentiation in vitro[7,9], limiting their applicability and usefulness for chronic studies.

When extracted from the beating adult heart, cardiomyocytes retain their structural and functional properties and have been widely used for acute experiments in every field of cardiovascular biology. However, within hours from isolation, cardiomyocytes undergo a process of structural and functional remodelling. This process results in the rapid loss of the highly specialised and sophisticated structure/function relationships (including altered cardiomyocyte morphology, the loss of t-tubules and sarcomeric disassembly[10,11]), that characterise beating, adult cardiomyocytes[12,13]. While they remain bona fide cardiac muscle cells, their use as a representative model becomes debatable, with significant implications for the relevance of medium-term/chronic studies.

During postnatal development, cardiomyocytes undergo maturation, which requires several weeks and results in the highly specialised and effective cells found in the adult heart. This involves a series of events that are similar, but opposite in direction, to those observed during culture of adult cardiomyocytes[14,15]. In addition, isolated foetal or neonatal cardiomyocytes fail to develop mature adult cardiac features in culture, a fate similar to that of pluripotent stem cell-derived cardiomyocytes[16]. However, when pluripotent stem cell-derived cardiomyocytes are implanted into adult hearts, they do undergo a progressive but incomplete phenotypic maturation[17,18]. These observations, coupled with widely reported evidence of rapid cardiomyocyte plasticity to physiological/pathological stimuli in vivo, support the concept of cardiomyocyte phenotypic adaptability and represent incontrovertible evidence that the in vivo environment is essential for the development and maintenance of specialised adult cardiac properties.

The precise factors responsible for the development and maintenance of the adult cardiac phenotype, including those lacking in culture, are unknown. While the regulation of cardiac phenotype is almost certainly multifactorial, there are three major aspects that are likely implicated. The first is the presence of adequate humoral stimulation and availability of metabolic substrates. Several culture protocols have been developed for isolated cardiomyocytes, particularly for gene transfections, but with limited success[19]. The second aspect is the presence of multicellularity and the extracellular matrix (ECM). Cell–cell interactions, via direct contact and/or paracrine secretions, contribute to cardiomyocyte function and regulation[20-22]. The ECM provides transmembrane regulation and controls membrane stiffness, again with important functional consequences[23]. However, cardiac multicellular preparations, including myocardial slices, suffer from similar culture-induced changes[7,9]. The final aspect, and one that is specifically associated with cardiac muscle behaviour, is the relentless cyclic electromechanical stimulation experienced by the myocardium. Electrical stimulation exerts positive effects but, in isolation, despite reducing cardiomyocyte remodelling, is still insufficient[24]. In addition, neonatal and pluripotent stem cell-derived cardiomyocytes possess intrinsic pacemaker activity and are continuously stimulated, yet they remain poorly differentiated. The factor that is absent in culture and may play a major role in regulating cardiomyocyte structure and function is sustained mechanical load. We and others have shown that structure and function of cardiomyocytes in vivo are modulated by changes in load, that the changes are graded and that they are reversible[25-28]. Chronic mechanical unloading of the heart induces atrophic remodelling, despite an unchanged humoral milieu, multicellularity, ECM and electrical stimulation[29]. This is not surprising, given that heart function reacts to changes in mechanical load and that a profound increase in haemodynamic load is essential for postnatal cardiac development[30].

In the beating heart in vivo, mechanical load plays an essential role in the regulation of acute and chronic cardiac performance. Despite the known interplay between mechanical load and cardiac function, the prolonged effects of physiological mechanical load on adult cardiac tissue in vitro remain under-investigated. A recent publication has highlighted the importance of continuous electromechanical stimulation for the long-term culture of myocardial slices[31]. However, only a single preload was tested, and myocardial slices were electrically stimulated at substantially subphysiological rates (0.2 Hz). A detailed characterisation of the structure, function and transcriptional properties of cultured myocardial slices was not conducted and the mechanisms regulating the preservation/dedifferentiation of myocardial slices in vitro remain unresolved. We investigated the role of biomimetic electromechanical stimulation with physiological preload to regulate the adult cardiac phenotype of myocardial slices in vitro and the gene set expression patterns involved. We used an inexpensive, simple and user-friendly platform that allows the maintenance and manipulation of adult myocardial slices in vitro. We and others observe significant dedifferentiation of myocardial slices produced from all species, with the fastest rates observed in slices produced from small mammals, such as rats. We therefore performed this study with freshly prepared rat myocardial slices and subjected them to different electromechanical stimulation protocols for 24 h. Optimised protocols were then applied to human heart failure myocardial slices for 24 h and healthy rabbit myocardial slices for 5 days (120 h).

Here, we report that a preload equivalent to sarcomere length (SL) = 2.2 μm is optimal for the maintenance of rat myocardial slice structural, functional and transcriptional properties at 24 h. Rat myocardial slices cultured at SL = 2.2 μm for 24 h have significant enrichment of gene sets associated with cardiac structure/function and transcription/translation, while gene sets associated with inflammation/cytokine signalling and extracellular matrix organisation are significantly downregulated. This results in the optimal maintenance of maximum contractility, $Ca^{2+}$ handling and t-tubule structure, which is likely mediated via the MLP-TCAP-Titin stretch-sensor complex. When cultured at other SLs for 24 h, rat myocardial slices display adaptive responses

compatible with dedifferentiation (structure, function and gene set expression patterns) without changes in viability or energy status. The maximum contractility of human heart failure myocardial slices cultured for 24 h is also optimally maintained at $SL = 2.2\,\mu m$. When cultured at $SL = 2.2\,\mu m$ for 5 days (120 h), the contractility of the rabbit myocardial slices remains stable throughout. This approach is a step towards a chronically stable in vitro preparation, which is a significant unmet need in cardiovascular science.

## Results

**Application of electromechanical stimulation to rat myocardial slices**. Rat myocardial slices were stretched uniaxially to sarcomere lengths (SL) within the physiological range ($SL = 1.8$–$2.4\,\mu m$) whilst field stimulated in culture. Electromechanically stimulated slices were compared with fresh myocardial slices (0 h) and with the most common slice culture technique, where slices are cultured without electromechanical stimulation (unloaded and non-stimulated) on a liquid–air interface (unloaded)[7,9]. To quantify the tissue stretch required to differentially preload slices, laser diffraction was used to assess the average slice sarcomere length (Fig. 1a). The aligned portion of each rat myocardial slice was trimmed and attached to custom-made 3D-printed rigid rectangular rings using surgical glue (Fig. 1c). The resting length of the tissue was then measured using calipers (green rectangle—Fig. 1c) and the rings were then mounted on the posts of custom-made stainless-steel stretchers (Fig. 1d). A 633-nm laser was directed at the slice surface and slices were progressively stretched until a diffraction pattern equivalent to $SL = 2.0\,\mu m$ was generated. The % stretch (distance between stretcher posts/resting length of myocardial slice × 100) was then calculated for $SL = 2.0$–$2.4\,\mu m$ at 0.1-$\mu m$ intervals. As $SL < 2.0\,\mu m$ could not be measured using this technique, a linear regression was used to calculate the % stretch required for $SL = 1.8$ (Fig. 1b). Rat myocardial slices were cultured at four SLs ($SL = 1.8, 2.0, 2.2,$ and $2.4\,\mu m$) to determine the optimal preload to maintain the adult cardiac phenotype in vitro (Fig. 1d). Stretched slices were cultured in a bespoke chamber and slices were continuously superfused with oxygenated culture media and electrically field stimulated with carbon electrodes (Fig. 1e). Unloaded slices were cultured on a liquid–air interface, as previously described[7,9] (Fig. 1f). Immunohistochemical staining and confocal microscopy confirmed that preload applied to slices was maintained in culture (Supplementary Fig. 1).

**Structure–function relationships: contractility, $Ca^{2+}$ handling and T-tubular structure in rat myocardial slices cultured for 24 h**. To determine the effect of prolonged electromechanical stimulation on rat myocardial slices, several structural and functional assays were performed. The maximum contractility (maximum force generation normalised to myocardial slice cross-sectional area) was measured using a force transducer after the slices had been cultured for 24 h and then removed from their stretchers. Rat myocardial slices cultured at $SL = 2.2\,\mu m$ had the best-preserved contractility, with only a small non-significant reduction compared with 0 h. This was significantly greater than unloaded and $SL = 2.0$-$\mu m$ myocardial slices (Fig. 2a, b). We also found that slices cultured unloaded on stretchers with electrical stimulation had a significant and rapid reduction in contractility in vitro (Supplementary fig. 2). To determine if the reduction in contractility was due to cardiomyocyte loss or structural/functional dedifferentiation of the tissue, whole myocardial slice viability was assessed using the CellTiter assay (Promega). There were no significant differences between the viability of fresh and cultured myocardial slices (Fig. 2c), indicating that alterations in

contractility were adaptations to different preload and culture conditions (similar finding was also observed with a Live/Dead assay — Supplementary fig. 3). The loss of contractility of unloaded slices corresponded with downregulation of many sarcomeric genes, while electromechanically stimulated slices ($SL = 1.8\,\mu m$ and $SL = 2.2\,\mu m$) had a gene expression profile more similar to 0 h slices (Fig. 2k). Of all the sarcomeric genes, expression of *CSRP3* (muscle LIM protein) was significantly higher in slices cultured at $SL = 2.2\,\mu m$, compared with both $SL = 1.8\,\mu m$ and unloaded (p-adjusted value = 0.09). $Ca^{2+}$ handling of myocardial slices was assessed with Fluo-4-AM and optical mapping (Fig. 2d). Slices cultured at $SL = 2.2\,\mu m$ had significantly larger $Ca^{2+}$ transient amplitude compared with unloaded slices (Fig. 2e), as well as a significantly faster rate of $Ca^{2+}$ rise (Fig. 2f) and significantly faster rate of $Ca^{2+}$ decay (Fig. 2g). Slices cultured at $SL = 2.2\,\mu m$ also had a significantly faster rate of $Ca^{2+}$ decay compared with slices cultured at $SL = 1.8$ and $2.4\,\mu m$. The time course of the $Ca^{2+}$ transient from slices cultured at $SL = 2.2\,\mu m$ was generally faster compared with other groups, but this did not reach statistical significance (Supplementary Fig. 4). Many genes associated with cardiac excitation–contraction coupling were downregulated in unloaded slices (Fig. 2l). The t-tubular structure of cultured myocardial slices was assessed with caveolin 3 staining and confocal microscopy (Fig. 2h). We found that slices cultured at $SL = 2.2\,\mu m$ had a similar t-tubule density compared with 0 h slices and that this was significantly greater than myocardial slices cultured at low mechanical loads (unloaded and $SL = 1.8\,\mu m$) (Fig. 2i). Slices cultured at $SL = 2.2\,\mu m$ also had a similar t-tubular regularity compared with 0 h slices and this was also significantly greater than unloaded slices (Fig. 2j). Genes associated with t-tubule stability were assessed and it was found that *TCAP* expression was increased in electromechanically stimulated slices and *junctophilin 2* and other t-tubule stability genes were reduced in unloaded slices (Fig. 2m).

**Conduction, connexins, β-adrenergic stimulation and arrhythmogenesis in rat myocardial slices cultured for 24 h**. The conduction velocity of rat myocardial slices cultured for 24 h was assessed using a multielectrode array system as previously described[4]. Conduction velocity was measured both longitudinal (Fig. 3a) and transverse (Fig. 3b) to myocardial slice myofibril direction. We found that slices cultured in unloaded conditions had a significant loss of longitudinal conduction velocity compared with 0 h slices and that this was also significantly reduced compared with $SL = 2.2\,\mu m$ (Fig. 3c). Due to a substantial increase in transverse conduction velocity in unloaded slices (Supplementary fig. 5A), the conduction anisotropy ratio was significantly reduced compared with both 0 h and $SL = 2.2\,\mu m$ (Fig. 3d). To determine if changes in conduction were due to changes in connexin 43 (Cx43) expression and localisation, immunohistochemical staining was used to label Cx43 and slices were imaged using confocal microscopy. We found that the area stained for Cx43 was maintained when rat myocardial slices were cultured with electromechanical stimulation ($SL = 1.8$–$2.2\,\mu m$) but was significantly reduced in unloaded slices (Fig. 3f). Unloaded slices and those cultured at $SL = 2.4\,\mu m$ also had a significant increase in Cx43 heterogeneity compared with 0 h slices (Supplementary Fig. 5B), but no changes in Cx43 lateralisation were found (Supplementary fig. 5C). *GJA1* expression was not reduced in any condition and there was no clear pattern in genes associated with Cx43 regulation and localisation at the intercalated disc (Fig. 3g). To determine if different culture conditions associated with alterations in conduction velocity and Cx43 expression led to a proarrhythmogenic substrate, rat myocardial slices were treated with increasing concentrations of isoproterenol. As expected, increasing concentrations of

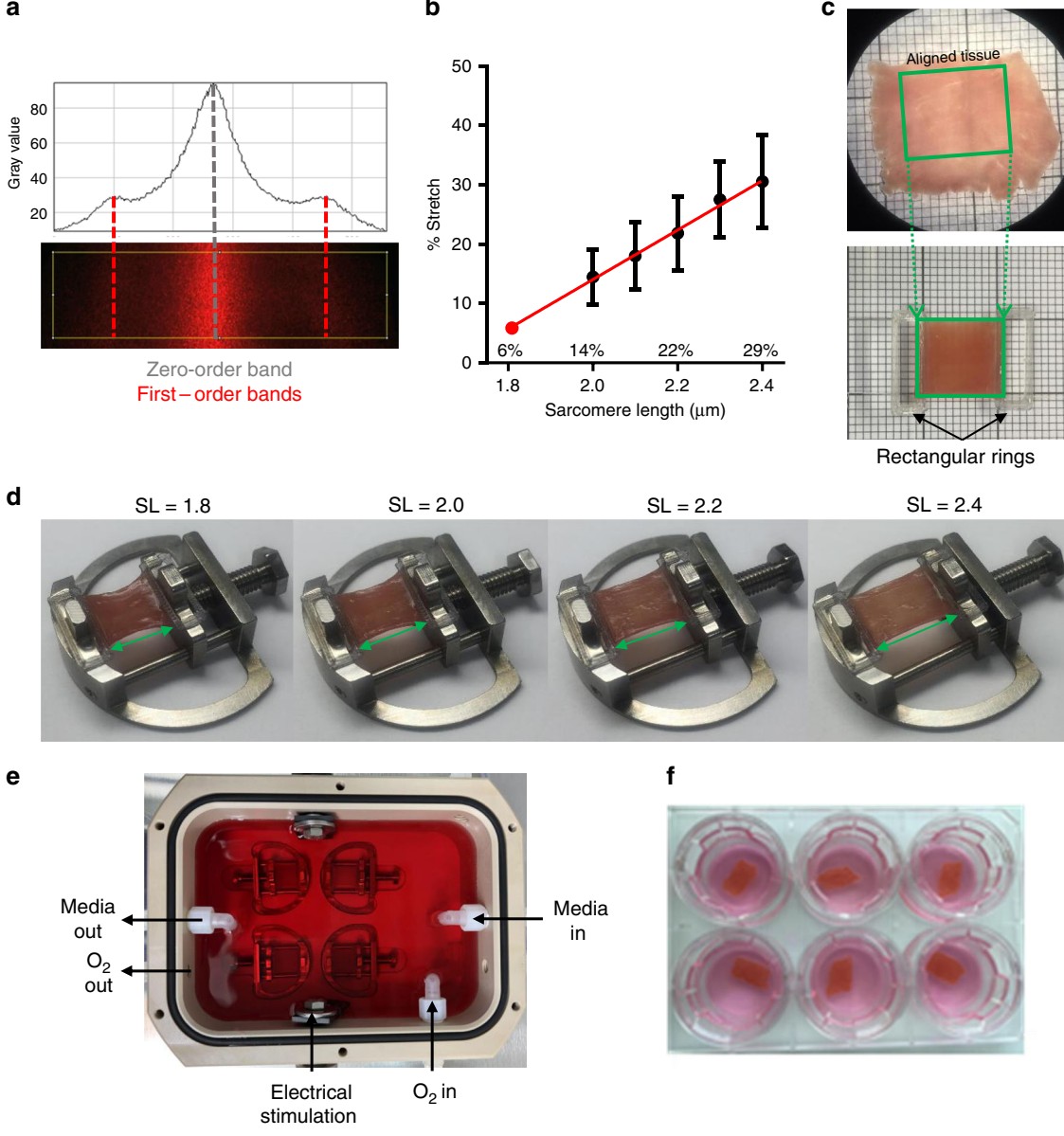

**Fig. 1** Application of electromechanical stimulation to rat myocardial slices. **a** Assessment of laser diffraction pattern. Peaks correspond to diffraction bands —the bright, central band corresponds to a zero-order band (grey), while the smaller bands on the left and right correspond to the less intense first-order bands (red). The distance between the zero-order and first-order band can be measured and used to calculate sarcomere length. **b** Percentage stretch required to set the average diastolic rat myocardial slice sarcomere length. Rat myocardial slices were progressively stretched until a diffraction pattern equivalent to SL = 2.0 μm was achieved. The % stretch was then measured using calipers. This was repeated at 0.1-μm intervals until SL = 2.4 μm. A linear regression was used to estimate SL ($r^2 = 0.4776$, $y = 41.67 \times -69.26$) (SL = 2.0 $N = 11$, SL = 2.1 $N = 16$, SL = 2.2 $N = 14$, SL = 2.3 $N = 15$ and SL = 2.4 $N = 12$). **c** Top—rat myocardial slice visualised using a macroscope. The slice is placed on a mm grid and the green rectangle highlights the aligned portion of the myocardial slice. Bottom—custom-made 3D-printed plastic rectangular rings are attached to opposite ends of the aligned portion of the myocardial slice using surgical glue. Rings are attached perpendicular to myofibril orientation. **d** Myocardial slice attached to the posts of a custom-made stretcher using rings. Images show the different stretches required to achieve SL = 1.8–2.4 μm in rat myocardial slices. **e** Custom-made culture chamber. Myocardial slices are superfused with culture media. Media was oxygenated directly in the culture chamber. Field stimulation was provided via carbon electrodes. **f** Six-well plate with Transwell inserts. Unloaded myocardial slices placed on a porous membrane and each well filled with 1 mL of culture media. N = number of myocardial slices. Mean ± standard error is shown on graphs. Source data are provided as a Source Data file

isoproterenol decreased time to peak force and had lusitropic effects (time to 50% force decay and time to 90% force decay in all conditions —Supplementary Fig. 6A–C). Slices cultured at SL ≥ 2.0 μm had a similar β-adrenergic response (similar EC$_{50}$) to 0 h slices (Supplementary Fig. 6D). However, it was impossible to measure the full β-adrenergic response of unloaded or SL = 1.8-μm slices, as they developed sustained tachyarrhythmia at very low concentrations of isoproterenol (Fig. 3h). The mechanism of

arrhythmia entry was typically an increase in aftercontractions, followed by tachyarrhythmia that was sustained when point stimulation was stopped (Fig. 3k). Unloaded and SL = 1.8-μm slices had substantially higher rates of aftercontractions (Fig. 3i) and significantly elevated arrhythmogenicity scores (Fig. 3j). Analysis of genes involved in β-adrenergic signalling did not explain the functional changes observed in response to isoproterenol stimulation (Fig. 3l). This makes it more likely that the changes

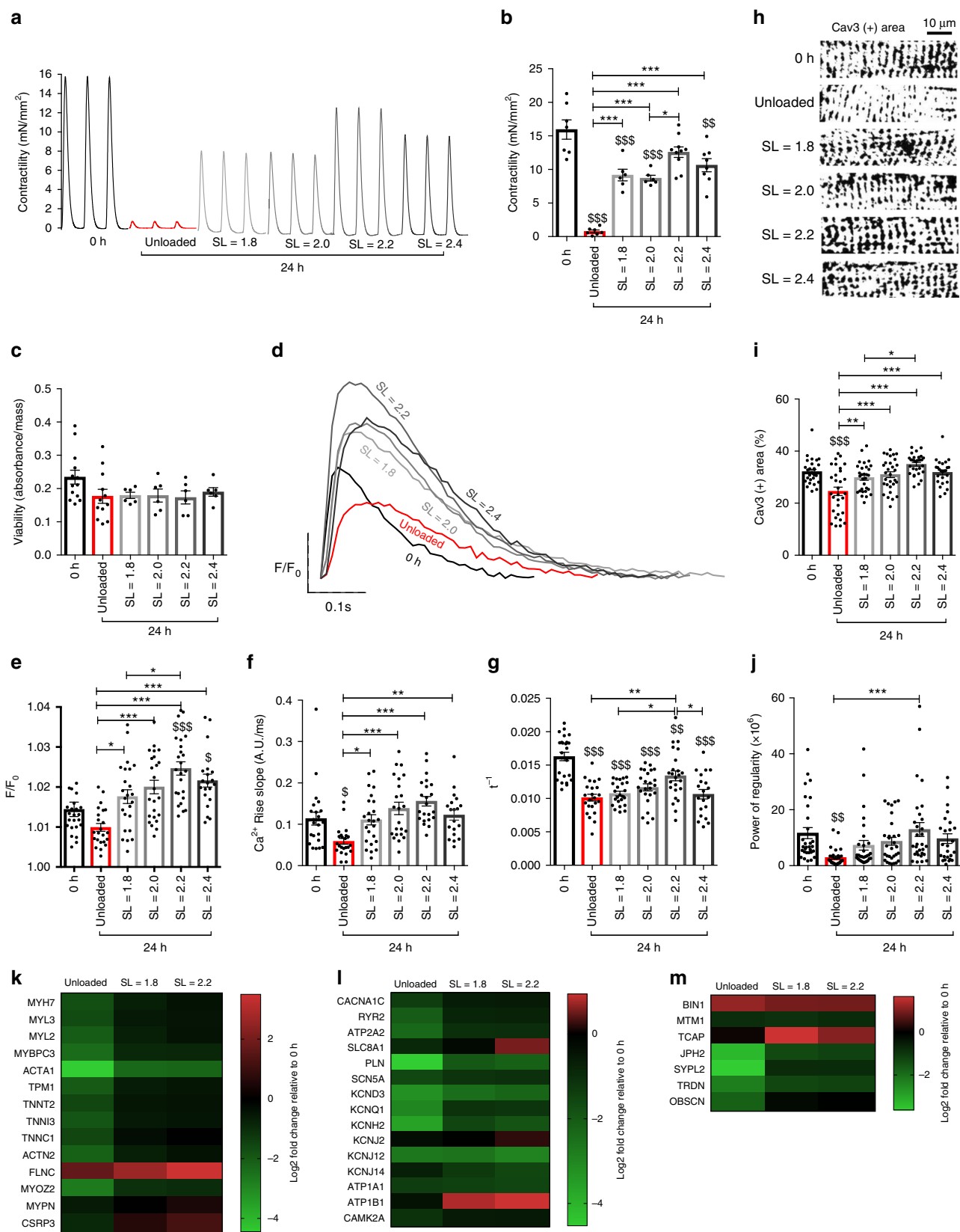

observed are due to post-translational modifications or altered phosphorylation status.

**Cell morphology, mitochondria and metabolism in rat myocardial slices cultured for 24 h.** The effects of electromechanical

stimulation on rat myocardial slice cell morphology following 24-h culture were investigated. The average cardiomyocyte area was measured using the caveolin 3 membrane stainings (Fig. 4a). Average cardiomyocyte area was maintained at SL ≥ 2.2 μm, while cardiomyocytes in unloaded slices developed significant atrophy

**Fig. 2** Structure–function relationships: contractility, $Ca^{2+}$ handling and T-tubular structure in rat myocardial slices cultured for 24 h. **a** Representative traces of rat myocardial slice contraction in all conditions. **b** Contractility of rat myocardial slices cultured for 24 h (0 h and unloaded $N = 7$, SL = 1.8 and 2.0 $N = 6$, SL = 2.2 $N = 10$ and SL = 2.4 $N = 8$). **c** Viability of rat myocardial slices cultured for 24 h assessed via CellTitre96 assay (0 h $N = 13$, unloaded $N = 12$ and SL = 1.8–2.4 $N = 6$). **d** Representative $Ca^{2+}$ transients acquired with Fluo-4 loading and optical mapping. **e** $Ca^{2+}$ transient amplitude of rat myocardial slices cultured for 24 h (0 h, SL = 1.8, 2.0 and 2.2 $N = 24/6$, unloaded $N = 23/6$, SL = 2.4 $N = 20/6$). **f** Rate of $Ca^{2+}$ rise of rat myocardial slices cultured for 24 h (0 h, unloaded and SL = 2.0 $N = 23/6$, SL = 1.8 $N = 24/6$, SL = 2.2 $N = 22/6$ and SL = 2.4 $N = 19/6$). **g** Rate of $Ca^{2+}$ decay of rat myocardial slices cultured for 24 h (0 h, SL = 1.8, 2.0 and 2.2 $N = 24/6$, unloaded $N = 23/6$ and SL = 2.4 $N = 20/6$). **h** Representative caveolin 3 staining of rat myocardial slices cultured for 24 h. **i** T-tubule density of rat myocardial slices cultured for 24 h (0 h, unloaded, SL = 1.8–2.2 $N = 30/6$, SL = 2.4 $N = 25/6$). **j** T-tubule power of the regularity of rat myocardial slices cultured for 24 h (0 h, unloaded, SL = 1.8–2.2 $N = 30/6$, SL = 2.4 $N = 25/6$). **k** Expression of genes associated with the sarcomere ($N = 3$). **l** Expression of genes associated with excitation–contraction coupling ($N = 3$). **m** Expression of genes associated with t-tubule stability ($N = 3$). Gene expression data from rat myocardial slices cultured for 24 h and expressed relative to 0 h rat myocardial slice gene expression. N = number of myocardial slices. For $Ca^{2+}$-handling data, N = number of regions analysed/number of myocardial slices. For t-tubule analysis, N = number of cardiomyocytes analysed/number of myocardial slices. Black dots represent individual data points. Mean ± standard error is shown on graphs. One-way analysis of variance (ANOVA) was used to determine whether there were any statistically significant differences between the means of groups. \$, \$\$, \$\$\$ = $p$ value < 0.05, 0.01 and 0.001, respectively compared with 0 h. *, **, *** = $p$ value < 0.05, 0.01 and 0.001, respectively, between the two groups highlighted by a bar. Source data are provided as a Source Data file

(Fig. 4b). The tightly defined architecture of the myocardium was also disrupted, with unloaded slices displaying a significantly reduced cardiomyocyte length:width ratio compared with 0 h and SL = 1.8–2.4 μm (Fig. 4c). Transmission electron microscopy studies confirmed the architecture of adult myocardium and the ultrastructure of cardiomyocytes was preserved when slices were cultured with electromechanical stimulation (Supplementary Fig. 7).

Our group has previously shown that stromal cells proliferate significantly on the surface of unloaded dog and human HF myocardial slices within 3 and 7 days, respectively[9]. Stromal cells were labelled with vimentin and myocardial slices were imaged using confocal microscopy. We found that there was a significantly faster and larger proliferation of these cells (48%) on the surface of unloaded rat myocardial slices at 24 h (Fig. 4d, e). The stromal cell population remained quiescent in all electromechanically stimulated slices. It is likely that these proliferating cells are cardiac fibroblasts, as a number of genes used to define this specific cell population[32] were more highly expressed in unloaded slices compared with SL = 1.8 and 2.2 μm (Fig. 4f).

It is well established that energy status is one of the key drivers of cardiac tissue differentiation, with increased energy production and efficient energy utilisation associated with the adult cardiac phenotype[33]. To determine if the structural and functional plasticity displayed by myocardial slices in culture was driven by altered mitochondrial structure and function, mitochondrial density was assessed using TOM20 staining and confocal microscopy, and metabolite concentrations were measured using high-performance liquid chromatography. We found no differences in mitochondrial localisation, with the mitochondria positioned at the A-band in all conditions (Fig. 4g). No differences in mitochondrial density were observed (Fig. 4h). However, slices cultured in unloaded conditions had a significantly reduced ATP concentration, while ATP concentration was maintained in electromechanically stimulated slices (Fig. 4i). Increasing mechanical load (SL ≥ 2.0 μm) preserved the ATP:ADP ratio of myocardial slices compared with 0 h slices (Fig. 4j). AMP, NAD, NADH, creatine and phosphocreatine concentrations were also assessed (Supplementary Fig. 8). The reduced ATP concentration in unloaded slices may explain at least some of the structural and functional remodelling we have observed. However, as functional (contractility and $Ca^{2+}$ handling) and structural (t-tubular structure) differences occur in electromechanically stimulated slices cultured at different SLs in the absence of changes in ATP concentration, it is likely that mechanisms not related to cardiac energy status are involved in the modulation of myocardial slice phenotype in vitro.

**Electromechanical regulation of gene expression in rat myocardial slices cultured for 24 h.** To investigate the effects of electromechanical stimulation on the transcriptome, RNA sequencing was performed. Rat myocardial slices cultured for 24 h were used. We chose to sequence four of our conditions (0 h, unloaded, SL = 1.8 μm and SL = 2.2 μm) to answer two specific questions: 1) what are the genes involved in dedifferentiation of cardiac tissue in vitro (0 h vs. unloaded)? and 2) what genes are involved in the regulation of cardiac structural and functional plasticity with different preloads (SL = 1.8 μm vs. SL = 2.2 μm)? When we compared the unloaded slices to 0 h slices (Fig. 5a), we found a strong enrichment of gene sets associated with re-entry into the cell cycle. An increased inflammatory response, with enrichment of cytokine and chemokine signalling was also found. Interestingly, TGF-β signalling and endothelial–mesenchymal transition gene sets were also enriched and may explain the observed increase in stromal cell proliferation. There was also significant down regulation of gene sets associated with energy production (cellular respiration, TCA cycle and oxidative phosphorylation) to suggest that unloaded slices may be reacting to hypoxia. Many gene sets associated with cardiac structure and function were downregulated. This included gene sets associated with cardiac muscle contraction, $Ca^{2+}$ release, conduction and cardiomyocyte repolarisation, the A and I-bands, t-tubules, the intercalated disc and the extracellular matrix.

Our structural and functional data indicate that providing a preload of SL = 2.2 μm significantly improves the structure and function of myocardial slices compared with SL = 1.8 μm and is beneficial for the maintenance of the adult cardiac phenotype in vitro. RNA sequencing provided further evidence of this at the transcriptome level (Fig. 5b). Slices cultured at SL = 2.2 μm had enrichment of cardiac muscle contraction and energy production gene sets. Gene sets associated with inflammation and cytokine/chemokine signalling were relatively downregulated, while those associated with transcription, translation and protein turnover were enriched. There also appeared to be less reorganisation of the extracellular matrix, including a downregulation of genes involved in collagen formation. MLP was most highly expressed at SL = 2.2 μm and there was significantly higher TCAP expression in electromechanically stimulated slices compared with unloaded. This indicates that the mechanotransduction of physiological electromechanical stimulation is mediated via the MLP-TCAP-Titin stretch sensor and that this is essential for the maintenance of the adult cardiac phenotype in vitro. As expected and when compared with 0 h, slices cultured at SL = 2.2 μm did appear to suffer some degree of dedifferentiation at the transcriptome level. However, when interpreted in the context

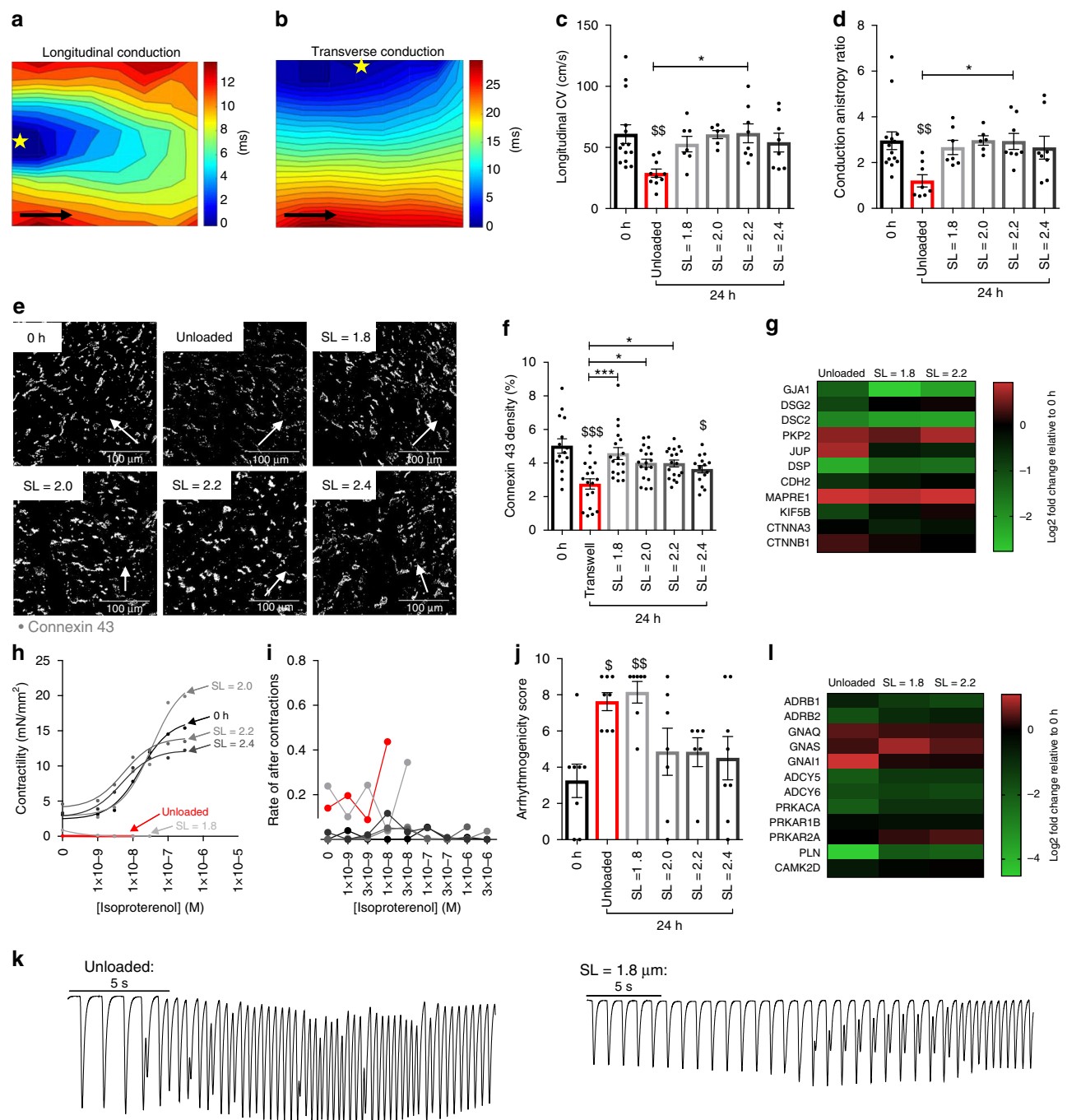

**Fig. 3** Conduction, connexins, β-adrenergic stimulation and arrhythmogenesis in rat myocardial slices cultured for 24 h. **a, b** Representative images of longitudinal and transverse conduction maps of a 0 h rat myocardial slice. Black arrows show myocardial fibre direction. Yellow star = location of the stimulating electrode. Area of conduction map: 4.9 × 4.9 cm. **c** Longitudinal conduction velocity of rat myocardial slices cultured for 24 h (0 h N = 14, unloaded N = 10, SL = 1.8 N = 7, SL = 2.0 N = 6, SL = 2.2 and SL = 2.4 N = 8). **d** Conduction anisotropy ratio of rat myocardial slices cultured for 24 h (0 h N = 14, unloaded, SL = 2.2 and 2.4 N = 8, SL = 1.8 N = 7 and SL = 2.0 N = 6). **e** Representative images of rat myocardial slices stained for connexin 43. White arrow shows the longitudinal axis of cardiomyocytes. **f** Connexin 43 density in rat myocardial slices cultured for 24 h (0 h and SL = 2.4 N = 15/6, unloaded, SL = 1.8–2.2 N = 18/6). **g** Expression of genes associated with connexin 43 regulation and localisation at the intercalated disc (N = 3).
**h** Isoproterenol dose response of rat myocardial slices cultured for 24 h (0 h, SL = 2.0–2.4 μm N = 7, unloaded N = 4 and SL = 1.8 μm N = 5). **i** Rate of aftercontractions with increasing concentration of isoproterenol of rat slices cultured for 24 h (0 h N = 8, unloaded, SL = 1.8–2.2 N = 6 and SL = 2.4 N = 7).
**j** Arrhythmogenicity score of rat myocardial slices cultured for 24 h (0 h, unloaded, SL = 2.4 N = 8, SL = 1.8 and 2.0 N = 7 and SL = 2.2 N = 6).
**k** Representative trace of an unloaded and SL = 1.8-μm rat myocardial slice developing a sustained tachyarrhythmia. **l** Expression of genes associated with the β-adrenergic stimulation (N = 3). Gene expression data from rat myocardial slices cultured for 24 h and expressed relative to 0 h rat myocardial slice gene expression N = number of myocardial slices. For Cx43 analysis, N = number of regions analysed/number of myocardial slices. Black dots represent individual data points. Mean ± standard error is shown on graphs. One-way analysis of variance (ANOVA) was used to determine whether there were any statistically significant differences between the means of groups. $, $$, $$$ = p value < 0.05, 0.01 and 0.001, respectively, compared with 0 h. *, **, *** = p value < 0.05, 0.01 and 0.001, respectively, between the two groups highlighted by a bar. Source data are provided as a Source Data file

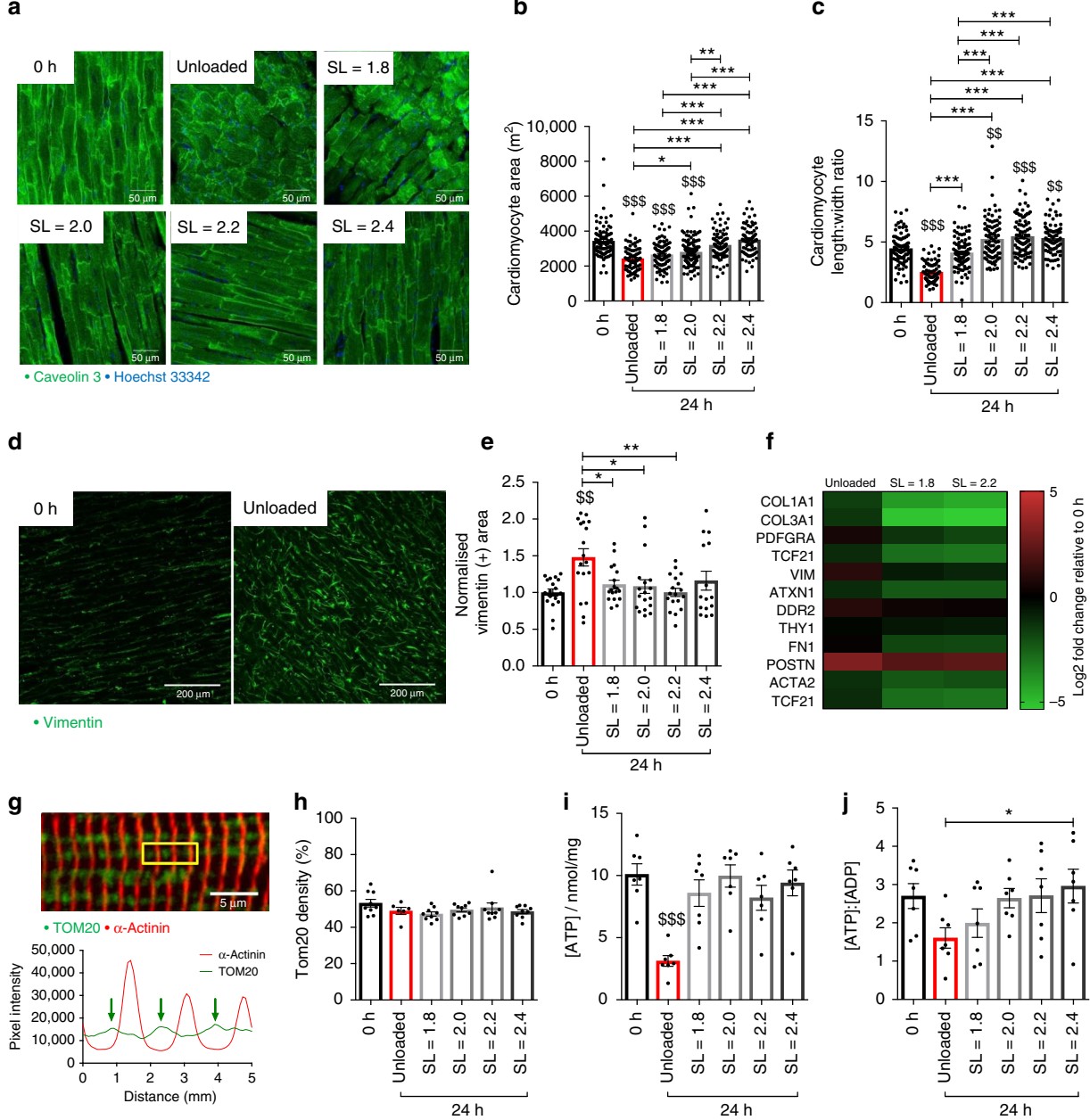

**Fig. 4** Cell morphology, mitochondria and metabolism in rat myocardial slices cultured for 24 h. **a** Representative images of the morphology of rat myocardial slices cultured for 24 h. Scale bar = 50 μm. **b** Average cardiomyocyte area (0 h, SL = 1.8 and 2.2 N = 90/6, unloaded N = 76/6, SL = 2.0 N = 135/6 and SL = 2.4 N = 75/6). **c** Cardiomyocyte length-to-width ratio of rat myocardial slices cultured for 24 h (0 h, SL = 1.8–2.2 N = 90/6, unloaded N = 76/6 and SL = 2.4 N = 75/6). **d** Representative images of cardiac fibroblasts on the surface of rat myocardial slices cultured for 24 h. Scale bar = 250 μm. **e** Quantification of stromal cell proliferation on rat myocardial slices cultured for 24 h (values are expressed as normalised vimentin + area of rat myocardial slices cultured for 24 h) (N = 18/6, except SL = 2.4 N = 15/6). **f** Expression of genes associated with cardiac fibroblasts and myofibroblasts (N = 3). **g** Top—representative image of a rat myocardial slice stained with TOM20 (mitochondria) and α-actinin (sarcomeric apparatus). Scale bar = 5 μm. Bottom—profile plot of dual staining. **h** TOM20 density of rat myocardial slices cultured for 24 h (N = 9, except unloaded N = 6). **i** Concentration of adenosine triphosphate (ATP) of rat myocardial slices cultured for 24 h (N = 7). **j** ATP:ADP ratio of rat myocardial slices cultured for 24 h (N = 7). Gene expression data from rat myocardial slices cultured for 24 h and expressed relative to 0 h rat myocardial slice gene expression. N = number of myocardial slices. For cardiomyocyte morphology analysis, N = number of cardiomyocytes analysed/number of myocardial slices. Black dots represent individual data points. Mean ± standard error is shown on graphs. One-way analysis of variance (ANOVA) was used to determine whether there were any statistically significant differences between the means of groups. \$, \$\$, \$\$\$ = p value < 0.05, 0.01 and 0.001, respectively, compared with 0 h. *, **, *** = p value < 0.05, 0.01 and 0.001, respectively, between the two groups highlighted by a bar. Source data are provided as a Source Data file

of our structural and functional findings, the process of dedifferentiation is significantly delayed when slices are cultured with biomimetic electromechanical stimulation with a physiological preload (SL = 2.2 μm).

**Electromechanical stimulation of human myocardial slices for 24 h and chronic culture of rabbit myocardial slices for 5 days (120 h).** The use of human tissue in vitro is vital for translational research. Our group received explanted human heart failure left

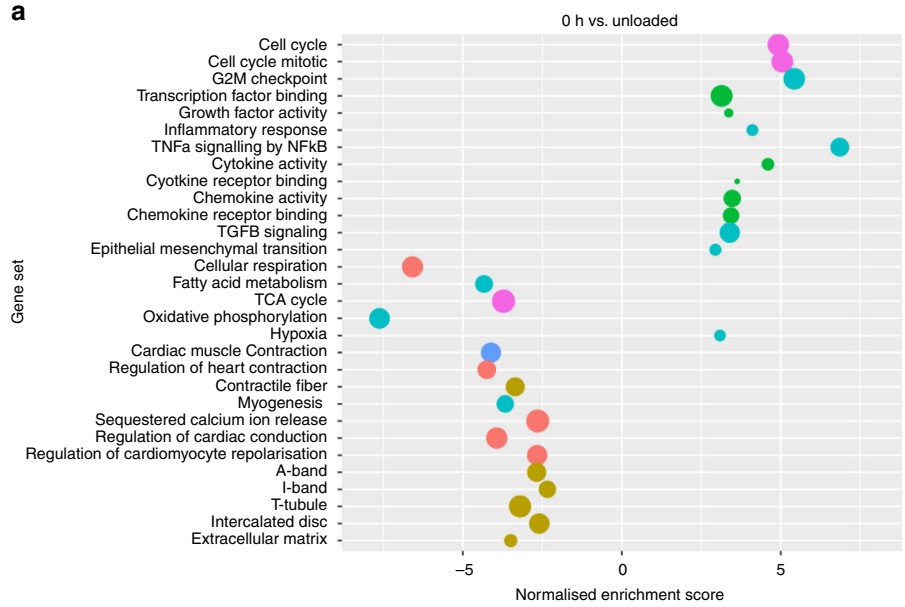

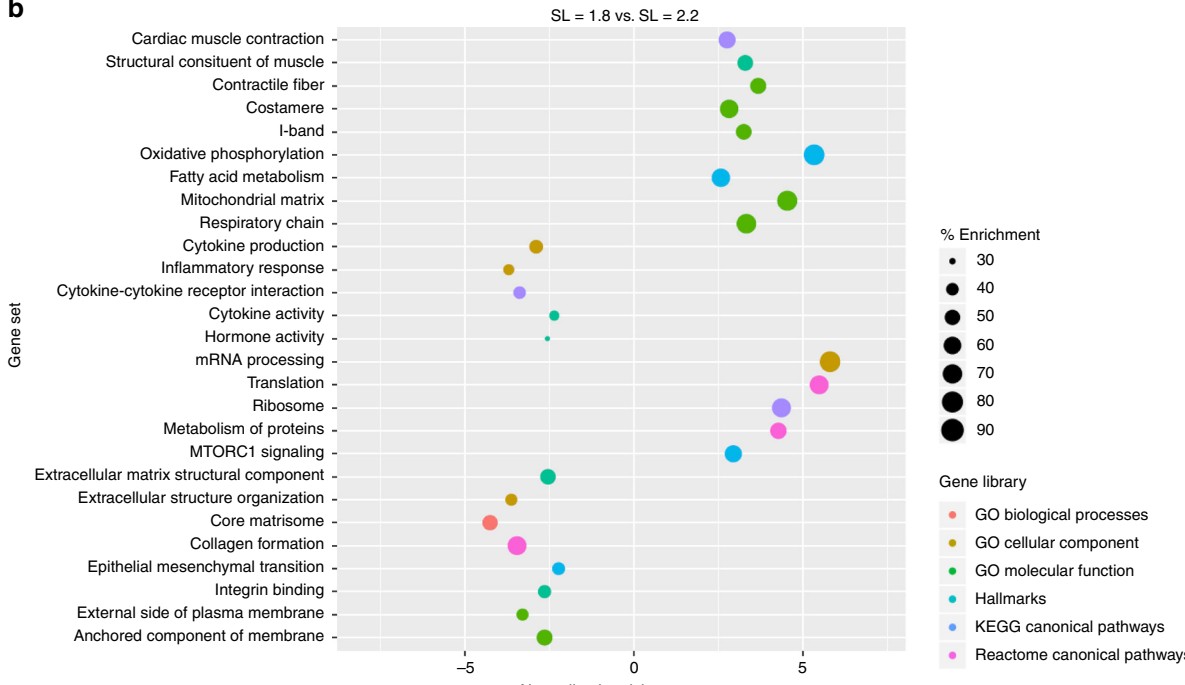

**Fig. 5** Electromechanical regulation of gene expression in rat myocardial slices cultured for 24 h. **a** Gene set enrichment analysis for unloaded rat myocardial slices compared with 0 h. A positive normalised enrichment scores equate to an enrichment of that gene set in unloaded slices. Gene sets on the Y axis, normalised enrichment score on the X axis, size of each point correlates with % enrichment (% of genes in a gene set that are enriched) and the colour of the point correlates with the gene set source ($N = 3$). **b** Gene set enrichment analysis for SL = 2.2 μm vs. SL = 1.8-μm rat myocardial slices. A positive normalised enrichment score equates to an enrichment of that gene set in SL = 2.2-μm slices. Gene sets on the Y axis, normalised enrichment score on the X axis, size of each point correlates with % enrichment (% of genes in a gene set that are enriched) and the colour of the point correlates with the gene set source ($N = 3$). SL = sarcomere length, GO = gene ontology, KEGG = Kyoto Encyclopaedia of Genes and Genomes. Source data are provided as a Source Data file

ventricular specimens, collected during cardiac transplantation. We applied the same electromechanical stimulation with physiological preload to human heart failure (HF) slices and measured their contractility after 24 h. A very similar trend to that seen in rat myocardial slices was observed, with human slices cultured at SL = 2.2 μm having the best-preserved contractility at 24 h (Fig. 6a). This was significantly better that the current published technique for culturing human myocardial slices (unloaded)[7,9].

Myocardial slices are an amenable model for culture, as their thinness (300 μm) facilitates the diffusion of $O_2$ and other essential nutrients. However, as we have previously shown, culturing myocardial slices using currently published techniques (liquid–air interface—unloaded) results in the rapid loss of contractile function amongst other structural/functional parameters, suggesting that this methodology is not representative of active, working cardiac tissue. Prolonged studies were performed

to see if the improvements in myocardial slice structure and function when cultured at SL = 2.2 μm could be extended to 5 days (Fig. 6b). Rabbit myocardial slices were used, as rabbit cardiac tissue is known to have similar cardiac properties to humans, particularly in terms of electrophysiology[34]. Rabbit slices were cultured at SL = 2.2 μm. We found that both the maximum (measured at 0, 24 and 120 h) and basal (measured every 2 h) contractility of rabbit slices cultured at SL = 2.2 μm increased over the first 24–48 h, as the tissue adapted to the in vitro environment and completely recovered from slicing (including the high [K$^+$] used for cardioplegic arrest). From 48 to 120 h, the tissue remained in a steady state, with only a subtle decrease in function. At 5 days, both the basal and maximum contractility of rabbit myocardial slices were almost completely preserved compared with 0 h (Fig. 6b).

## Discussion

Our data demonstrate that the application of electromechanical stimulation with physiological preload significantly improves the preservation of myocardial structure and function in culture and provides a step towards a stable in vitro model of the adult myocardium. More specifically, we have demonstrated that the application of a preload equivalent to SL = 2.2 μm significantly reduces culture-induced remodelling compared with other pre-loads within the physiological range or unloaded. We have demonstrated that almost all structural and functional parameters of the cardiac phenotype are regulated by changes in mechanical load in vitro and that the regulation of this phenotypic plasticity is multifactorial.

The ability to maintain myocardial slices in a functional state in vitro, which is not possible with current published techniques, provides a significantly improved and much needed platform for cardiovascular research. The specific benefits of this improved model include prolonged preservation of the adult cardiac phenotype in vitro, intact native cardiac multicellularity and tissue architecture, acquisition of high-quality sensitive data on cardiac structure and function in real time and an ability to investigate representative responses to pharmacological and mechanical load stimuli in vitro. In vitro adult cardiac models undergo a rapid process of tissue dedifferentiation and loss of the adult cardiac properties (phenotype and function) in culture. Our group and others have previously shown that this also occurs when myocardial slices are cultured in unloaded conditions[7,9]. We have demonstrated that the loss of the adult cardiac phenotype can be significantly delayed in culture, with rabbit myocardial slices remaining highly functional for 5 days. Further to Fischer et al.[31], we have demonstrated that mechanical load must be applied within a very specific biomimetic range (~SL = 2.2 μm) in order to be reproducible and prevent culture-induced dedifferentiation. Myocardial slices have preserved cardiac multicellularity and the extracellular matrix, uniquely allowing the study of all cardiac cell populations in a representative environment. This is fundamental to a deeper understanding of cardiac physiology, as the non-myocyte and non-cellular components of the myocardium play vital roles in the regulation of cardiomyocyte structure and function[20]. In this study, we confirmed that electromechanical stimulation not only preserves cardiomyocyte function, but also maintains non-myocyte populations in a physiological state. We have confirmed that electromechanical stimulation prevents the massive proliferation of cardiac fibroblasts in vitro previously observed[9], while also maintaining a more physiological gene expression profile. The ultrathin nature of myocardial slices combines the benefits of a complex multicellular 3D preparation with the simplicity of acquiring structural and functional data from a 2D monolayer. In this study, we have fully characterised

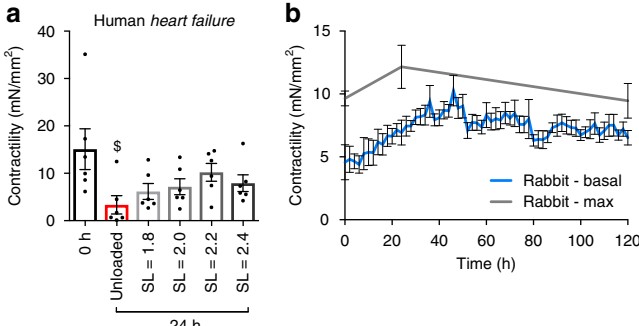

**Fig. 6** Electromechanical stimulation of human myocardial slices for 24 h and chronic culture of rabbit myocardial slices for 5 days (120 h). **a** Contractility of human heart failure myocardial slices cultured for 24 h ($N$ = 6). **b** Contractility of rabbit myocardial slices cultured for 5 days—0 h ($N$ = 6), 24 h ($N$ = 11) and 120 h ($N$ = 6) (grey) and the basal contractility was measured every 2 h during the 5-day experiments ($N$ = 6) (blue). N = number of myocardial slices. Black dots represent individual data points. Mean ± standard error is shown on graphs. One-way analysis of variance (ANOVA) was used to determine whether there were any statistically significant differences between the means of groups. $, $$, $$$ = p value < 0.05, 0.01 and 0.001, respectively compared with 0 h. *, **, *** = p value < 0.05, 0.01 and 0.001, respectively, between the two groups highlighted by a bar. Source data are provided as a Source Data file

both the structure and function of myocardial slices in culture using several high-sensitivity assays. We have measured direct force generation and contractile reserve, the most relevant para-meters of cardiac function, and have demonstrated that both remain stable for 5 days. In addition, we have demonstrated that cultured (24 h) myocardial slices responded to isoproterenol sti-mulation in a comparable manner to gold-standard drug-screening models[35,36], with both 0 h and cultured myocardial slices (SL = 1.8–2.4 μm) displaying an EC$_{50}$ in the nanomolar (×10$^{-8}$ M) range. The model can also be used to investigate car-diac remodelling in response to both unloading and overload, which we have sensitively quantified using laser diffraction.

Myocardial slices display remarkable plasticity to changes in preload, with changes occurring at the structural, functional and transcriptional level within 24 h. Our data suggest that these changes are adaptive, as the viability assays performed did not demonstrate significant changes in whole-slice viability and gene sets associated with apoptosis/necrosis were not enriched in any condition. Multiple mechanisms can detect changes in mechan-ical load[37]. The primary function of the sarcomere is to generate force; however, several of its molecules have been shown to have load sensing and mechanotransduction capabilities. Titin and muscle LIM protein (MLP) have been previously implicated in mechanotransduction and hypertrophy signalling pathways[38,39]. In our study, the MLP-TCAP-Titin complex was upregulated with electromechanical stimulation and physiological preload, particularly at SL = 2.2 μm, where MLP (*CSRP3* gene) had sig-nificantly higher expression compared with both SL = 1.8 μm and unloaded slices. *TCAP* was also more highly expressed in both SL = 1.8 μm and SL = 2.2-μm slices. Costameres are known to be important in the lateral transmission of force[40] and genes asso-ciated with them were also significantly enriched in slices cultured at SL = 2.2 μm compared with SL = 1.8 μm. Our data support the existence of multiple mechanisms to regulate load-induced changes in cardiac phenotype.

When conducting prolonged in vitro experiments with myo-cardial slices, several considerations should be made. Several species differences exist in terms of cardiac function[34]. We found that slices produced from small mammals, including rats,

dedifferentiated at faster rates than large animals due to their higher metabolic rates[41]. Rat myocardial slices were useful in identifying optimal electromechanical conditions for prolonged culture (SL = 2.2 μm) and are adequate for 24-h experiments. The preparation of mouse myocardial slices is challenging due to the small and curved geometry of the ventricle. This results in an inability to isolate discrete sheets of myocardial fibres, resulting in reduced viability[3]. Larger mammal slices have a lower metabolic rate and subsequent rate of dedifferentiation, making them more amenable to prolonged experiments. Rabbit myocardial slices were cultured at SL = 2.2 μm with only a very subtle decline in contractility observed over 5 days. Real-time monitoring of myocardial slice force generation was used for 5-day experiments to confirm the stability of the preparation in vitro. Human cardiac tissue is of great value to translational studies, and in this study, we were able to optimally maintain human HF myocardial slices at SL = 2.2 μm for 24 h. It is likely that human myocardial slices can be cultured for substantially longer periods, but in this study, we focussed on the prolonged culture of healthy tissue, as there is often significant variability in the pathology and quality of human HF specimens. Future studies applying experimental translational strategies to human myocardial slices cultured for prolonged periods should be conducted, but this was outside of the remit of this study.

While the application of physiological electromechanical stimulation significantly improves the maintenance of the mature adult cardiac phenotype in vitro, we still observed some dedifferentiation occurring. In our study, myocardial slices were preloaded at physiological sarcomere lengths. However, the mechanical load experienced by the myocardium in vivo consists of both preload and afterload. We focussed on preload as it directly influences the contractility of the myocardium via the Frank–Starling relationship[42], and the actual application of physiological afterload is technically very challenging and has only been achieved for acute studies so far. Another aspect that can be improved is the humoral stimulation of the slice in culture. The media used in this study did not contain hormones, including noradrenaline and thyroid hormones, which are known to be essential for cardiac structural and function remodelling[43–45]. In addition, the media used lacked binding proteins. Finally, we utilised field stimulation to excite the myocardial slices and physiological stimulation may be better recreated using a point/line stimulation. It was possible to point stimulate slices, as in our arrhythmia provocation studies, but practically it is not as robust as field stimulation for prolonged use in sterile conditions. Despite these limitations, our data show substantial improvements in myocardial slice structure and function following the application of electromechanical stimulation with physiological preload. It is likely that optimisation and application of the variables previously mentioned will lead to further small and incremental improvements in cultured myocardial slices, as we move closer to recreating the physiological in vivo environment. Further experiments with longer time points are now required to confirm the suitability of the approach for chronic experiments with rat and human heart failure tissues.

We have developed a platform to culture and manipulate living, beating adult myocardial slices in vitro and have demonstrated that electromechanical stimulation with specific physiological preload is required for the maintenance of adult cardiac tissue in vitro. This is a significant advance compared with current gold-standard cardiac models. We have shown that electromechanical stimulation affects structure, function and many gene set expression circuits, which regulate many aspects of the cardiac phenotype, including sarcomeres, $Ca^{2+}$ handling and t-tubules. This approach represents a significant step towards the unmet need of a stable adult cardiac in vitro preparation, which may open avenues in the testing of pharmacological agents and experimental therapeutic approaches.

## Methods

**Animal experiments**. All animal experiments complied with institutional and national regulations. Our use of living cardiac tissue was approved by Imperial College London. The procedures we describe here were performed under license by the UK Home Office, in accordance with the United Kingdom Animals (Scientific Procedures) Act 1986. Animals were killed following guidelines established by the European Directive on the protection of animals used for scientific purposes (2010/63/EU).

**Preparation of rat myocardial slices**. Male *Sprague Dawley* rats, with a body mass between 300 and 500 g, were sedated with isoflurane (4% isoflurane and 4 L/min $O_2$) and sacrificed using cervical dislocation and dissection of the carotid arteries.

(1) The heart and lungs were rapidly removed from the chest and placed in warm (37 °C) heparinized Tyrode's solution (see Table 1) + 30 mM 2,3-Butanedione Monoxime (BDM, Acros Organics) for 5–10 s to remove blood from the ventricle.
(2) The heart and lungs were then placed in cold (4 °C) Tyrode's solution + BDM and transferred to the laboratory.
(3) A left ventricular tissue block was prepared by dissecting off the lungs and right ventricle. An incision was made down the interventricular septum and the left ventricle was opened and flattened.
(4) The tissue block was then mounted on an agarose-coated specimen holder using surgical glue (Histoacryl, Braun) epicardial surface facing down.
(5) The specimen holder was placed in the bath of a high-precision vibratome (7000smz-2, Campden Instruments) submerged in oxygenated cold (4 °C) Tyrode's solution + BDM.
(6) The tissue block was sliced using a ceramic blade, longitudinal to myocardial fibre orientation and from the endocardium down to the epicardium. The vibratome was set to vibrate at 80 Hz, with an amplitude of 2 mm and a Z-axis error < 1.0 μm.
(7) Once cut and prior to being used in acute studies or being put in culture, slices were kept in oxygenated cold (4 °C) Tyrode's solution + BDM under mesh holders.

**Preparation of rabbit myocardial slices**. To preserve viability, rabbit hearts require in situ cardioplegic arrest.

(1) Rabbits (Male, *New Zealand White*, 2.5–3.0 kg obtained from Envigo Laboratories, Huntington, UK) were sedated with ketamine (15 mg/kg) and medetomidine hydrochloride (0.25 mg/kg), intubated with a 3-mm endo-tracheal tube and hearts explanted under terminal anaesthesia (using mechanical ventilation with isoflurane).
(2) An abdominal laparotomy was performed and the abdominal aorta and inferior vena cava visualised by blunt dissection.
(3) Slow retrograde infusion (over 3–5 min) of 150–200 mL of cold (4 °C) heparinised (500 units/kg) cardioplegia solution (Harefield Hospital formulation, Terumo BCT, UK) via a 20 G Leadercath Arterial catheter (Vygon, UK) into the ascending aorta was used to arrest the heart. Just prior to infusion, the inferior vena cava and abdominal aorta distal to the catheter were clamped, and an incision made in the inferior vena cava (proximal to the venous clamp) to allow blood to expel during cardioplegia administration.
(4) The heart was then rapidly excised and immersed in cold (4 °C) cardioplegia solution and transferred to the laboratory.
(5) The tissue block was prepared and sliced in the same manner as for rats (see *Preparation of Rat myocardial slices*); however, low calcium (0.9 mM $CaCl_2$) oxygenated cold (4 °C) Tyrode's solution + 30 mM BDM was used.
(6) Prior to recordings, slices were incubated for 20 min, at room temperature, in normal Tyrode's solution (1.8 mM $CaCl_2$— see Table 1) + 30 mM BDM to re-equilibrate the calcium concentration in the slice.

**Preparation of human HF myocardial slices**. Human HF samples were provided by the NIHR Cardiovascular Biomedical Research Unit at the Royal Brompton and Harefield NHS Foundation Trust and Imperial College London. The study performed conformed to the principles outlined in the Declaration of Helsinki and the investigation was approved by a UK institutional ethics committee (NRES ethics number for biobank samples: 09/H0504/104 + 5; Biobank approval number: NP001-06-2015 and MED_CT_17_079) and Imperial College London. Informed consent was obtained from each patient involved in this study.

(1) Human hearts (end-stage heart failure) were perfused with cold (4 °C) cardioplegia solution and arrested in situ prior to being explanted. The inferior 1/3rd of explanted hearts were obtained from patients undergoing cardiac transplantation.

| Table 1 Preparation of Tyrode's solution | | |
|---|---|---|
| **Solid reagents** | | |
| | **Mass (g)** | **Final concentration (mM)** |
| 2,3-Butanedione Monoxime | 3.00 | 30.00 |
| Glucose | 1.86 | 1.00 |
| HEPES | 2.38 | 10.00 |
| Potassium chloride | 0.45 | 6.00 |
| | 0.34 | 4.50 |
| Sodium chloride | 8.18 | 140.00 |
| **Solutions** | | |
| | **Volume (ml)** | **Final concentration (mM)** |
| 1 M magnesium chloride | 1.00 | 1.00 |
| 1 M calcium chloride | 1.80 | 1.80 |

Add the following to 1 L of distilled water. Only add 2,3-butanedione monoxime and 6 mM potassium chloride to slicing solution. A volume of 4.5 mM potassium chloride is used for making physiological recordings

(2) The specimens were placed in a 1-L container of cold cardioplegia, placed on ice and transported to the laboratory.

(3) In order to prepare a left ventricular tissue block, the left ventricle of the specimen was identified and isolated from the rest of the tissue. A 1.5-cm$^2$ tissue block was dissected out of the left ventricular free wall by making an incision through the full thickness of the ventricular wall using a scalpel.

(4) The tissue was mounted epicardial surface down and sliced in the same manner as both rat and rabbit myocardial slices.

**Twenty-four hour myocardial slice culture and quantification of preload.**

(1) Only aligned areas, where parallel myofibrils run in the same direction, were used in this study. A macroscope was used to assess global myofibril orientation and the slice was then trimmed using a razor blade to isolate the most highly aligned area (Fig. 1c).

(2) Custom-made rings were printed in biocompatible T-glase plastic (taulman3D, USA) using an Ultimaker Original + 3D printer (Ultimaker, Netherlands). The rings were then attached to opposite ends of the slice, perpendicular to myofibril direction, using surgical glue (Histoacryl, Braun—Fig. 1c—bottom).

(3) The length and width of the unloaded slice were then measured using calipers.

Custom-made stretchers were used to apply uniaxial preload to the slice (Fig. 1d). The stretchers consisted of a fixed and moveable post, with the distance between the posts regulated by mechanically turning a screw mechanism.

(4) Using tweezers, one of the rings was gently lifted to allow the slice to hang freely. The free-hanging ring was then draped over the fixed post and the ring held by the tweezers was carefully placed over the moveable post (it was imperative that myocardial slices were not overstretched or mechanically damaged during this process).

(5) Each myocardial slice was then stretched to the desired average tissue SL by turning the screw mechanism.

(6) Once the desired preload had been applied, the myocardial slice stretcher was placed into the culture chamber.

The preload applied to myocardial slices was quantified using laser diffraction.

(1) Aligned myocardial slices were mounted on stretchers, as described above, and were placed in a glass Petri dish filled with Tyrode's solution on a micromanipulator.

(2) The beam of a 6333-nm helium–neon laser (Lasos, Germany) was directed at the slice surface and the position of the slice was adjusted laterally, until a clear diffraction pattern was visualised.

(3) The laser was positioned 2 cm above the slice surface and the diffraction pattern was projected on a grid 20.5 cm below.

(4) The diffraction pattern was captured digitally using a camera (Logitech C920 HD, Switzerland) and the distance between the zero-order and first-order bands was calculated using ImageJ (National Institute of Health, USA). This distance was used to calculate sarcomere length using trigonometry[46].

(5) Diffraction patterns could not be achieved for SL < 2.0 μm. We measured the % stretch (distance between two post/ myocardial slice resting length × 100) for SL = 2.0–2.4 μm at 0.1-μm intervals.

(6) A linear regression was used to calculate the % stretch required for SL = 1.8 μm.

The % stretch calculation takes both the slice length and a small portion of the ring into account, as the distance between the posts was measured, instead of the tissue length, for practicality issues. This explains why a 6% stretch is required for SL = 1.8 μm, despite the resting SL being ≈1.8 μm. Slices were stretched to the desired average slice sarcomere length, measured using calipers, directly prior to being put in culture.

**Twenty-four hour culture of myocardial slices in culture chambers.** In order to apply electromechanical stimulation and physiological preload to rat and human HF myocardial slices in culture, myocardial slice stretchers (Fig. 1d) and custom-made chambers, in which four stretchers could be loaded (Fig. 1e), were used.

(1) Myocardial slice stretchers, culture chambers and tools required for putting slices in culture were autoclaved prior to experiments.

(2) During myocardial slice preparation, culture chambers were assembled in a sterile laminar flow cabinet.

(3) Superfusion tubing and air filters were connected to the chamber and the chamber was filled with 250 mL of culture medium (medium-199 (Sigma, USA) + 0.1% ITS (containing 1.0 mg/mL bovine insulin, 0.55 mg/mL human transferrin and 0.5 μg/mL sodium selenite at 100× concentration) (Sigma, USA) + 3% Penicillin/Streptomycin (Sigma, USA)).

(4) The culture medium was continuously bubbled with 95% $O_2$/5% $CO_2$ (BOC, UK) at 1 L/min and recirculated using a peristaltic pump at 15 mL/min.

(5) Immediately after each slice had been prepared, custom-made 3D-printed T-Glase rectangular rings were mounted at opposite ends of the aligned portion of a slice orthogonal to myocardial fibre direction, as described above ("Application of mechanical load—1 & 2").

(6) The length and width of each mounted myocardial slice were measured and recorded.

(7) Mounted myocardial slices were transferred to the sterile laminar flow cabinet.

(8) In total, 1.5 mL of cold Tyrode's solution was pipetted onto the myocardial slice, so that it was floating in the Petri dish.

(9) Using a pair of tweezers, myocardial slices were mounted on myocardial slice stretchers, as described above ("Application of mechanical load—3").

(10) Each myocardial slice was then stretched to the desired average tissue SL by turning the screw mechanism (SL = 1.8–2.4 μm).

(11) The myocardial slice stretcher was then carefully placed into the culture chamber and its position noted. It took less than 5 min to put each myocardial slice in culture.

(12) Once four myocardial slices had been placed in culture, the chamber was sealed.

(13) The culture chamber was transferred to an incubator and slices were cultured at 37 °C.

(14) Rat myocardial slices were stimulated at 1 Hz (width: 10 ms, voltage: 50 V) and human heart failure myocardial slices were stimulated at 0.5 Hz (width: 10 ms, voltage: 30–50 V). Due to the large culture medium volume, the actual voltage experienced by myocardial slices was equivalent to 5–10 V. The voltage used was equivalent to 200% of the voltage required to capture the slice and as voltage was increased recruitment was observed.

Electromechanically stimulated slices were compared with slices cultured in the absence of electromechanical stimulation or preload. Slices cultured in the absence of electromechanical stimulation were kept on a liquid–air interface using Transwell membranes. Slices were placed on the Transwell membranes and excess solution was removed from the slice surface and edges. One millilitre of culture medium was added to the well of a six-well plate. The cultures were kept in humidified air at 37 °C with 5% $CO_2$ in an incubator.

**Five-day culture of rabbit myocardial slices.** For cultures longer than 24 h, a Bose Electroforce 5100 was used. This device was able to provide a uniaxial mechanical load to myocardial slices, whilst continuously recording myocardial slice contractility. Continuous monitoring of myocardial slice performance was required in order to optimise culture conditions during prolonged cultures.

(1) The Electroforce culture chamber, superfusion tubing and medium bottle were autoclaved prior to experiments.

(2) The entire Electroforce system was assembled in an incubator and kept at 37 °C for the entire experiment.

(3) The force transducer, linear motor, superfusion and electrical stimulation were all tested prior to experiments.

(4) The medium bottle was filled with 1.5 L of culture medium and oxygenated with 95% $O_2$/5% $CO_2$ at 3 L/min.

(5) Rabbit myocardial slices were prepared, mounted and measured as described above ("Application of mechanical load—1,2,3").

(6) The mounted myocardial slice was then loaded into the culture chamber of the Electroforce system.

(7) The culture chamber was sealed and filled with culture medium using the peristaltic pump. The medium was recirculated at 15 mL/min throughout the culture.

(8)  The myocardial slice was field stimulated using bipolar stimulation with platinum electrodes at 1 Hz (width: 10 ms, voltage: 40 V).

(9)  The myocardial slice was slowly stretched over 30 min, until the slice was progressively stretched, until a preload equivalent to SL = 2.2 µm was applied (rabbit: 35% stretch—confirmed via laser diffraction, as described above—"Twenty-four hour myocardial slice culture and quantification of preload"). Myocardial slice length was digitally monitored using a camera and 1-mm graph paper attached to the back of the culture chamber throughout stretching.

(10)  Myocardial slices were cultured for 5 days and culture medium was completely replaced after 72 h.

(11)  During culture, the electrical stimulation had to be occasionally increased (width: 10–20 ms, voltage: 40–55 V), when contractile alternans occurred.

(12)  A 10-s recording of myocardial slice contractility was taken every 15 min for the duration of the experiment.

(13)  At 120 h, myocardial slice maximum contractility was measured by the Electroforce. Slices were stretched in a stepwise manner, until maximum isometric contraction was recorded. Myocardial slice contractility was normalised to slice width.

Rat myocardial slices cultured at SL = 2.2 µm using myocardial slice stretchers and the Bose Electroforce have a similar contractility at 24 h (Supplementary Fig. 9).

**Contractility**. Slice contractility was assessed using a force transducer (Harvard Apparatus, USA). To assess the contractility of fresh slices (0 h) and those cultured without electromechanical stimulation (unloaded), the areas with the most highly aligned myofibrils were selected and custom-made rings were attached as described above. Slices cultured with electromechanical stimulation (SL = 1.8, 2.0, 2.2 and 2.4 µm) were carefully removed from the culture chamber and then removed from stretchers. The slices were then attached to the force transducer using the attached rings. Rat myocardial slices were field stimulated at 1 Hz, 10 ms and 10–30 V and human HF and donor slices were field stimulated at 0.5 Hz, 10 ms and 10–30 V. The slices were continuously superfused in 37 °C oxygenated Tyrode's solution. The slices were progressively stretched in a stepwise manner, until maximum isometric contraction was obtained. Data were recorded using AxoScope software and peak amplitude analyses were conducted using Clampfit software (both Molecular Devices, USA). A video of the contraction of human HF and rat myocardial slices can be found here: https://images.nature.com/original/nature-assets/nprot/journal/v12/n12/extref/nprot.2017.139-sv8.mp4

**Ca²⁺ handling**. To assess $Ca^{2+}$ handling, myocardial slices were loaded with Fluo-4 AM (5.56 µg/mL) (Thermo Fisher Scientific, USA) in medium-199 and 0.001% Pluronic F-127 (Thermo Fisher Scientific, USA). Zero-hour and unloaded myocardial slices were loaded in a six-well dish with a circular gauze and washer placed over the slice to prevent contraction during loading. Electromechanically stimulated slices were loaded in a six-well dish and kept on their stretchers at their respective sarcomere lengths. Rat myocardial slices were loaded for 10 min at 37 °C and then directly transferred to the optical mapping setup. Slices were placed in a chamber and superfused with 37 °C Tyrode's solution + BDM. BDM is required to prevent movement artefact during calcium transient acquisition. Slices were field stimulated at 1 Hz, 10 ms and 20 V. Changes in Fluo-4 fluorescence were detected using an ORCA-Flash4.0 LT CMOS camera (Hamamatsu) at 154 frames/second and data were acquired using HCImage Live software (Hamamatsu). $Ca^{2+}$ transient amplitude, time to peak, time to 50% decay, time to 90% decay, rate of $Ca^{2+}$ rise and $Ca^{2+}$ decay rate were measured using Clampfit software (Molecular Devices). A video of the Fluo-4 signal recorded from the myocardial slice surface can be found here: https://images.nature.com/original/nature-assets/nprot/journal/v12/n12/extref/nprot.2017.139-sv9.mp4

**Viability**. Myocardial slice viability was assessed using two assays:

CellTiter 96 AQ_{ueous} One Solution Cell Proliferation Assay (Promega) Following culture, slices were removed from the stretchers and the rings were detached using a razor blade. For 0 h and unloaded slices, the aligned area was selected and used. Slices were blotted dry and their mass measured using an electronic balance. Slices were put into individual wells of a 96-well plate and 200-µl medium-199 + 40 µl of CellTiter 96 AQ_{ueous} One solution reagent were added to each well. The plate was incubated at 37 °C for 20 min in a humidified, 5% $CO_2$ atmosphere. Following incubation, the absorbance at 490 nm was immediately recorded using a 96-well plate reader (Labtech). The absorbance value for each slice was normalised to slice mass.

Live/Dead viability assay (Molecular Probes) and wide-field microscopy Following culture, slices were removed from the stretchers and the rings were detached using a razor blade. For 0 h and unloaded slices, the aligned area was selected and used. The stock reagents were diluted as per the manufacturer's instructions to produce a solution with 2 µM calcein-AM and 4 µM ethidium homodimer-1. Two millilitres of solution was added to each well of a six-well plate and slices were placed in the individual wells and covered with a circular gauze and washer. Slices were incubated at 37 °C for 45 min in a humidified, 5% $CO_2$ atmosphere. Following incubation, slices were washed in phosphate- buffered

solution (PBS) and then fixed in 4% formaldehyde solution for 15 min at room temperature. Circular gauzes and washers were again used to make sure that the slice was fixed as flat as possible. Tiled images of the whole myocardial slice were automatically collected using a ×10 objective on a Zeiss AxioObserver equipped with a motorised stage. To assess slice surface viability, we measured the percentage of area stained with fluorescent calcein-AM signal using ImageJ.

**Conduction velocity**. The conduction velocity of myocardial slices was assessed using a multielectrode array (MEA) system (Multichannel Systems), which allows noninvasive synchronous multifocal recording of extracellular field potentials. The MEA consists of 60 microelectrodes arranged in an $8 \times 8$ matrix, with a 100-µm electrode diameter and an inter-electrode distance of 700 µm, providing a recording area of about $4.9 \times 4.9$ mm². Slices were positioned over the microelectrode array and held in contact using a plastic slice holder. Slices were continuously superfused with oxygenated Tyrode's solution at 37 °C. Electrical stimulation (bipolar pulses, 1 Hz, 1 ms and 2 V) was applied via one of the MEA microelectrodes. Field potential data were acquired simultaneously from all 60 microelectrodes with a sampling frequency of 50 kHz. Slices were stimulated longitudinal and transverse to known myofibril direction. A custom MATLAB (MathWorks) software was used to locate field potentials from recording electrodes, produce activation maps and calculate conduction velocities, using time of maximum amplitude. A ratio of the longitudinal and transverse conduction velocities of each slice ($CV_L:CV_T$) was used as a measure of CV anisotropy.

**β-adrenergic response and arrhythmia provocation**. Isoproterenol stimulation was used to assess the susceptibility of myocardial slices to arrhythmia following culture. Slices were attached to a force transducer as described above (see the "Contractility" section) and a consistent preload was applied. Slices were point stimulated (1 Hz, 1 ms and 3 V) such that electrical propagation was longitudinal to fibre direction. Slices were superfused with oxygenated Tyrode's solution at 37 °C with the addition of $1 \times 10^{-9}$ to $3 \times 10^{-6}$ M (−)–isoproterenol hydrochloride (Sigma) in half-log increments + 0.1 mM ascorbic acid (Sigma). Isoproterenol concentrations were added cumulatively, once changes in contractility had stabilised following the administration of the previous concentration. Arrhythmia was assessed using two parameters, the rate of aftercontractions at each concentration of isoproterenol and the concentration of isoproterenol required to stimulate sustained tachyarrhythmia. The threshold for tachyarrhythmia was converted into a linear arrhythmogenicity score such that tachyarrhythmias triggered by lower concentrations of isoproterenol translated to a higher arrhythmogenicity score (see Table 2). The β-adrenergic response to isoproterenol was also analysed in each condition by measuring the amplitude, time to peak, time to 50% decay and time to 90% decay at each isoproterenol concentration using Clampfit software.

**Immunohistochemical staining and confocal microscopy**. Following culture, slices were washed in PBS and then fixed in 4% formaldehyde solution for 15 min at room temperature. The samples were not subjected to freezing or sectioning in order to prevent tissue damage and to preserve cardiac structure. Slices were permeabilised in 1% Triton X-100 for 1 h at room temperature. Slices were then blocked in 5% foetal bovine serum, 5% bovine serum albumin and 10% horse serum in PBS solution for 2 h. Slices were incubated in the primary antibody in PBS at 4 °C overnight, and then washed three times for 30 min. Slices were incubated in the secondary antibody in PBS for 3 h at room temperature and kept covered with aluminum foil to prevent excessive light exposure. Slices were then washed a final time and stored in PBS at 4 °C. See Table 3 for antibody concentrations. Immunolabelled slices were imaged using a confocal microscope (Zeiss LSM-870).

**High-performance liquid chromatography**. For analysis of ATP-related metabolites, cultured slices were quickly dried to culture media and were snap frozen in

---

**Table 2 Arrhythmogenicity score**

| Isoproterenol threshold (M) | Arrhythmogenicity score |
|---|---|
| 0 | 9 |
| $1 \times 10^{-9}$ | 8 |
| $3 \times 10^{-9}$ | 7 |
| $1 \times 10^{-8}$ | 6 |
| $3 \times 10^{-8}$ | 5 |
| $1 \times 10^{-7}$ | 4 |
| $3 \times 10^{-7}$ | 3 |
| $1 \times 10^{-6}$ | 2 |
| $3 \times 10^{-6}$ | 1 |
| No tachyarrhythmia | 0 |

The threshold for tachyarrhythmia was converted into a linear arrhythmogenicity score such that sustained tachyarrhythmia developed at lower concentrations of isoproterenol translated to a higher arrhythmogenicity score

## Table 3 Antibodies

**Antibodies**

| Antibody/stain | How it was created | Dilution | Manufacturer | Catalogue number |
|---|---|---|---|---|
| *Primary antibodies* | | | | |
| Caveolin 3 | Raised in mouse | 1:500 | BD Biosciences | 610421 |
| Connexin 43 | Raised in rabbit | 1:2000 | Sigma Aldrich | C6219 |
| a-Actinin | Raised in mouse | 1:2000 | Sigma Aldrich | A7811 |
| Vimentin | Raised in chicken | 1:3000 | Thermo Fisher Scientific | PA1-16759 |
| TOM20 | Raised in mouse | 1:750 | Santa Cruz | sc-17764 |
| Isolectin B4 | Biotin-conjugated | 1:1000 | Life Technologies | 121414 |
| *Secondary antibodies* | | | | |
| Alexa 488 | Raised in donkey, anti-mouse | 1:2000 | Life Technologies | A21202 |
| Alexa 546 | Raised in donkey, anti-rabbit | 1:2000 | Life Technologies | A10040 |
| Alexa 647 | Raised in goat, anti-chicken | 1:2000 | Life Technologies | A-21449 |
| *Nuclear staining* | | | | |
| Hoechst 33342 | | 1:1000 | Life Technologies | H3570 |

liquid nitrogen. Slices were freeze dried before extraction with 0.4 perchloric acid at a ratio of 1:100 using a glass homogeniser. Following centrifugation at 15115x*g* for 5 min at 4 °C, supernatants were neutralised with 2 M KOH. Subsequent centrifugation at 15115x*g* for 5 min at 4 °C removed precipitated potassium perchlorate and supernatants were used for analysis on the day of extraction. HPLC analysis of ATP, ADP, AMP, phosphocreatine, creatine, NAD and NADH (as adenosine diphosphoribose— ADPR) was carried out. The HPLC instrument was Agilent 1100 system with 150 × 4.6 mm, 3 μm BDS Hypersil column protected by Security Guard (Phenomenex).

**RNA isolation**. Following culture, myocardial slices were immediately frozen in liquid nitrogen and stored at −80 °C. A TissueLyser LT (Qiagen) was used to disrupt the sample. The frozen slices were transferred to a pre-cooled 2-mL round-bottom tube containing a 5-mm stainless-steel ball and 500 μl of TRIzol (Thermo Fisher Scientific). The TissueLyser was set to 40 Hz and left for 3–5 min, until the sample was completely lysed. One-hundred microlitres of chloroform (Sigma) was added to each tube and then vortexed for 15 s. The tube was then left in a rack for 5 min at room temperature and intermittently shaken. The tube was then centrifuged at 12879 × *g* at 4 °C for 15 min and phase separation occurred. The top aqueous layer (containing the RNA) was then removed. An equivalent volume (to that of the aqueous layer removed) of 70% ethanol was then added and the solution mixed via pipetting. All of the solution was then transferred to an RNeasy spin column (Qiagen) and the manufacturer's instructions were then followed. Following collection, RNA was frozen and kept at −80 °C.

**RNA sequencing**. TruSeq Stranded mRNA libraries were multiplexed and sequenced with the average of 70 million DNA fragments per sample (75 base pair paired-end reads). Quality control was performed using FastQC software (Andrews S (2010) FastQC: a quality control tool for high-throughput sequence data. Available online at: http://www.bioinformatics.babraham.ac.uk/projects/fastqc—version 0.11.2). Sequencing reads were aligned to Rattus Norvegicus (Rnor_6_0) reference genome by Tophat (version 2.0.10) using the set of known genes provided by Ensembl database (release 90) with the average alignment rate of 96%[47]. The raw number of read pairs mapped to each Ensemble gene was calculated with HTSeq (version 0.6.0) in 'union' mode[48]. Reads (or read pairs) that overlap more than one gene or mapped to multiple locations were discarded. The raw count data were normalised using DESeq2 (version 1.14.1)[49]. Differential gene expression analysis for various pairs of conditions was performed using DESeq2 (version 1.14.1). Genes were ranked according to their log2 fold change and a preranked gene set enrichment analysis was conducted using GSEA software from the Broad Institute (available at http://software.broadinstitute.org/gsea/index.jsp). Genes were compared with gene sets (H, C2 and C5) in the Molecular Signatures Database. We only present gene sets with a normalised enrichment score q value < 0.005 and that related to our structural and functional assays. Plots were produced in R using 'ggplot2'.

**Reporting summary**. Further information on research design is available in the Nature Research Reporting Summary linked to this article.

## Data availability

Source data for Figs. 1b, 2b, c, 2e–g, 2 i–m, 3c–d, 3f–j, 4b–c, 4e, f, 4h–j, 5a, b and 6a, b and Supplementary Figs. 1B, 2, 3, 4A–C, 5A–C, 6A–C, 8A–I, and 9 are provided with the paper. RNA-seq data have been deposited in the ArrayExpress database at EMBL-EBI (www.ebi.ac.uk/arrayexpress) under accession number E-MTAB-7842. Full microscopy image datasets are available from the corresponding author upon reasonable request.

## Code availability

The MATLAB code used to analyse conduction velocity data is available from the corresponding author upon reasonable request.

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

## Acknowledgements

We would like to acknowledge the work of Paul Brown, Andy Rochester (both Mechanical Instrumentation Workshop, Imperial College), Michael Lennon (Materials Workshop, Imperial College London) and Joel Thomas for the design and manufacture of both the myocardial slice stretchers and culture chambers used in this study, Felicity McGrath for her help optimising the laser diffraction studies, Stephen Rothery (Facility for Imaging by Light Microscopy, Imperial College London) for assistance with confocal and wide-field microscopy and Imperial BRC Genomics Facility for carrying out RNA sequencing. We thank the British Heart Foundation for funding this work through an MBPhD studentship to SW (FS/15/35/31529) and the BHF Centre for Regenerative Medicine (RM/17/1/33377). HPLC experiments were funded by the National Science Centre, Poland (NCN (Poland) Harmonia grant no: 2016/22/M/NZ4/00678). TEM was funded by MRC project grant number (MR/R010676/1) to AT. Human tissue samples were provided by the Cardiovascular Research Centre Biobank at the Royal Brompton and Harefield NHS Foundation Trust, UK (NRES ethics number for biobank samples: 09/H0504/104 + 5; Biobank approval number: NP001-06-2015 and MED_CT_17_079).

## Author contributions

S.A.W. performed experiments and wrote the paper. J.D., I.B. and M.Z. performed experiments. S.S.A. analysed RNA-Seq data. R.J.J. provided rabbit tissue. A.S. provided human tissue. A.T. performed transmission electron microscopy experiments. R.T.S. performed high-performance liquid chromatography experiments. S.E.H. reviewed the paper. FP performed experiments and directed the project. C.M.T. conceived and directed the project. All authors discussed the results and edited the paper.

## Additional information

**Competing interests:** The authors declare no competing interests.

