## [Peer Review File · Nature Communications]

Reviewers' comments:

Reviewer #1 (Remarks to the Author):

This research consists of taking rat myocardial slices, placing them on a stretching scaffold in culture, keeping them alive and healthy for 24 hours, and measuring sarcomere length. It is still in the preliminary stage, with the authors not having tested their hypothesis beyond the 24hr stage. Without any additional data to substantiate these claims, it is not possible to determine the long term effects or usefulness of their research in human specimens. Further experiments and data are necessary.

General Comments for Author:

The manuscript describes an improved method to preserve structural and functional integrity of **adult** cardiac tissue slices (also called: organotypic culture).

The **major finding** is that a preload equivalent to a sarcomere length of 2.2 μm is optimal for the maintenance of the tissue.

Several groups have tried to improve longevity of **adult** cardiac tissue slices before. While it is relatively easy to maintain embryonic or neonatal cardiac tissue viable for several days, adult cardiac tissue shows a rapid loss of viability and dedifferentiation, precluding chronic experiments. Therefore, experiments with **adult** cardiac tissue slices usually must be performed within hours or at least on the same day to be relevant.

In their manuscript, Watson et al. demonstrate structural and functional integrity for 24 hours. This is probably a further improvement and will ease **acute** experiments. However, it is still not enough to perform real **chronic** experiments. Several days would e.g. be necessary to analyse remodelling or to perform adenoviral-mediated gene transfer.

It is self-explanatory for an organ like the heart that “**biomimetic electromechanical stimulation**” is necessary for an advanced organotypic culture model of the heart and many other groups have already elaborated models to provide these physiological stimuli.

For this reason, the major contribution of this manuscript is the implementation of **sarcomere length measurements** and the thorough demonstration of the importance of this parameter to maintain the original differentiation of cardiac tissue using state of the art methods.

In contrast to many other groups that merely describe their model, we get a precise information on the optimal value for at least one parameter (sarcomere length). Though the result itself is not surprising, using it will help to optimise other groups' models as well and will be of interest to others in the community as well.

In summary, every aspect of this study has been addressed before:

- Methodology of generating and maintaining viable cardiac tissue slices
- Importance of preload for the preservation of cardiac tissue
- Measurements of sarcomere length *in vitro* and *in vivo*

However, given the importance of this area of research, the number of papers published, combining this information and the accurate demonstration that we have reliable research tools for further studies is very important.

Specific Comments for Author:

Title

My suggestion would be to talk about “tissue **slices**” instead of “issue” already in the title of the manuscript. In addition, the authors might think about focussing the title and the paper on “sarcomere length” and “preload”.

Introduction

Introduction – Line 78 & 109:

“With time, their morphology and function start to resemble those of developing cardiomyocytes found in the foetal and neonatal heart (14). “

“Chronic mechanical unloading of the heart induces a foetal phenotype, despite an unchanged humoral milieu, multicellularity, ECM and electrical stimulation (28). “

I would hesitate to compare dedifferentiated adult tissue slices with foetal or neonatal tissue slices. There are too many aspects that are different and primarily related to necrosis, apoptosis, fibroblast overgrowth, etc. .

The authors use too references to support their statements:

Reference 14:

Szibor M, Pöling J, Warnecke H, Kubin T, Braun T. Remodeling and dedifferentiation of adult cardiomyocytes during disease and regeneration. *Cell Mol Life Sci*. 2014;71(10):1907–16.

This work is a review paper that also addresses dedifferentiation in vitro but not dedifferentiation of cardiac tissue slices and, although some aspects (e.g. gene expression) resemble a foetal phenotype, it doesn't support the author's simplification.

Reference 28:

Razeghi P, Sharma S, Ying J, Li Y-P, Stepkowski S, Reid MB, et al. Atrophic remodeling of the heart in vivo simultaneously activates pathways of protein synthesis and degradation.

Circulation [Internet]. 2003

Nov 18 [cited 2018 Feb 2];108(20):2536–41. Available from:

<http://www.ncbi.nlm.nih.gov/pubmed/14610007>

This manuscript deals with atrophic remodeling induced by heterotopic transplantation of the rat heart and is suited to highlight the importance of loading conditions but not the development of a foetal phenotype.

Introduction – Line 87:

“However, when pluripotent stem cell-derived cardiomyocytes are implanted into adult hearts they do undergo a **phenotypic maturation** (18). “

Reference 18:

Chong JJH, Yang X, Don CW, Minami E, Liu Y-W, Weyers JJ, et al. Human embryonic-stem-cell-derived cardiomyocytes regenerate non-human primate hearts. *Nature* [Internet]. 2014 Jun 30 [cited 2018 Jan 7];510(7504):273–7. Available from:

<http://www.nature.com/articles/nature13233>

This article provides evidence for a “progressive but **incomplete** maturation over a three-month period” (cited from the abstract of the original article by Chong JJH et al.). The word ‘incomplete’ is important.

Results:

“Application of electromechanical stimulation to myocardial slices”

I would have considered the first section of the “Results” (lines 127-161) including Figure 1 to be a part of the Methods section.

“Structure-Function Relationships: Contractility, Ca²⁺ handling & T-Tubular Structure” (lines 162 ff. with Figure 2)

Line 164:

“The maximum contractility (maximum force generation normalised to myocardial slice cross-sectional area).”

The term “contractility” is used by the authors for the force generated normalised to myocardial slice cross-sectional area. The term “contractility” should be replaced.

Line 224 :

“T-tubule density, assessed by quantifying the density of caveolin-3 staining [...] Ultrastructure, including T-tubular structure should have been assessed by electron microscopy.”

“Conduction, Connexins, β -Adrenergic Stimulation & Arrhythmogenesis” (lines 234 ff. with Figure 3)

Line 235:

“The conduction velocity of cultured rat myocardial slices was assessed using a multielectrode array system as previously described (4).”

Reference (4): Camelliti P, Al-Saud SA, Smolenski RT, Al-Ayoubi S, Bussek A, Wettwer E, et al. Adult human heart slices are a multicellular system suitable for electrophysiological and pharmacological studies. J Mol Cell Cardiol [Internet]. 2011 Sep [cited 2015 Jul 1];51(3):

Unfortunately, there was an error while I was trying to create a session. Even if the method is not described thoroughly, it is important to describe the parameter that was used for the MEA maps (time of maximum amplitude [i.e. U_{max}], time of maximum increase of voltage [i.e. $\Delta U/\Delta t$]).

Figure 3 A/B:

Information on the size of the square (cm) shown and the fiber orientation (e.g. an arrow) would be useful as well as using the same scaling for Figure 3A and 3B.

Figure 3 E:

The images (pdf-file as well as WORD-file) provided to the reviewer do not provide enough resolution to judge the connexin 43 staining.

“Cell Morphology, Mitochondria & Metabolism”

(lines 280 ff. with Figure 4)

Line 291:

“It is likely these proliferating cells are cardiac fibroblasts [...]”

This is probably true but endothelial are probably also involved and could be easily stained e.g. with von Willebrand factor.

“Electromechanical regulation of gene expression”

(lines 328 ff. with Figure 5)

Annotation:

The authors demonstrate a nice preservation during the first 24 hours.

What would be important for future developments would have been an analysis of the changes thereafter. What are the changes that occurred despite preload and electrical stimulation after 24 hours in culture?

The authors are asked to comment on this topic.

It might even be important and helpful to provide some data on the unsolved problems.

Discussion:General comments:

The discussion is quite long, and many aspects are redundant. Therefore, it could be more concise.

The most important information that is provided by the experiments is simply:

“we have demonstrated that the application of a preload equivalent to $SL=2.2\mu m$ significantly reduces culture-induced remodelling compared to other preloads within the physiological range or unloaded.” (line 370)

The authors claim: “The ability to maintain myocardial slices in a steadier state in vitro, which is not possible with current published techniques, provides a new and much needed platform for cardiovascular research.” (lines 374-375)

The authors try to elaborate that their model is new. Definitely, it is not a new but an improved model.

Therefore, it would be helpful to point out what they think are the most beneficial changes as compared to existing models. They focus on preloading but this, too, has been used before. Since many other scientists have used modifications of the model before, it would be helpful to discuss the changes that have been made that they think were most important.

Another very important point would have been to discuss the temporal limitations:
When experiments are done with this experimental set-up:
What is the time frame for physiologically relevant studies?
The authors may comment on this topic more precisely.

When we have the choice between different species.
Which species seems to be best suited to get relevant data?

Could the authors comment on the use of their method in mice.

Although the discussion also mentions the limitations of the models, it would be helpful to separate the limitations more clearly (e.g. by a headline, underlining).

Methods

General comment:

Reproducibility of the model:

The authors describe an interesting further development of already existing tissue culture models and prove its relevance. For this reason, one of the most important topics is, that other researchers can reproduce the model. Therefore, a very precise and very detailed description of the model is mandatory. Although they might quote their previous work, other researchers should be able to use this technique with the help of this paper alone. Supplemental material might be helpful if the article would be too long otherwise.

Specific comments:

Methods – Line 470

“Rat myocardial slices - Rats were sedated [...] “

Please specify the rats used preferably for adult slices (age, species, body weight).

Methods – Line 484

“The tissue block was sliced using a ceramic blade, longitudinal to myocardial fibre orientation and from endocardium down to the epicardium. The vibratome was set to vibrate at 80Hz, with an amplitude of 2mm and a Z-axis error <1.0µm.”

The authors should comment on the blades used:

Did they observe an improvement using the (very expensive) ceramic blades as compared to (less expensive) metal blades? The authors should specify the ceramic blade that was used.

The fibre orientation changes from endocardium to epicardium.

Was the orientation of the tissue block changed during slicing?

Methods – Line 562

“Rat myocardial slices were field stimulated at 1Hz, 10ms, 10-30V and human HF & donor slices were field stimulated at 0.5Hz, 10ms, 10-30V.”

The authors should comment on the voltage used.

Did they determine a threshold and stimulate with e.g. 150% or 200% of the threshold?

Did the slice respond in an all-or-nothing behaviour or was recruitment and force development influenced by the voltage used?

What kind of stimulation did they use?

The authors should specify the stimulator and the impulse precisely.

Methods – Line 566

A video of the contraction of human HF 566 and rat myocardial slices can be found here:

<https://images.nature.com/original/nature-assets/nprot/journal/v12/n12/extref/nprot.2017.139-sv8.mp4>

This video is helpful for all researchers that would like to use this model.

The video would be even better if anode and cathode would be labelled.

Methods – Line 612

Slices were stimulated longitudinal and transverse to known myofibril direction. A custom MATLAB (MathWorks) software was used to locate field potentials from recording electrodes, produce activation maps and calculate conduction velocities as previously described(3,4).

Important issue:

Since the authors submit this manuscript to a high-ranked journal, I would have expected **ultrastructural images** as well. I would have considered this necessary to demonstrate structural integrity.

Minor Comments:

Abstract – Line 38:

The abbreviation „SL“ („sarcomere length“) should be defined in the abstract as well.

The authors should use one spelling for the word „caliper“ respectively „calliper“.

Figure 1: The authors use „gray“ in the figure and „grey“ in the figure legend.

21 December 2018

Dear Reviewers,

We have reflected on your initial comments and performed additional experiments to address your concerns. We have included a point-by-point reply and explanation of the additions and revisions below. The authors of the paper agree that these changes have considerably strengthened the manuscript.

In this manuscript we demonstrate that ultrathin slices of adult cardiac tissue can be maintained beating *in vitro* for 24 hours without a run down in their function. While several groups have reported maintained viability and some functional parameters in other *in vitro* myocardial models, to the best of our knowledge there are no published data to show preserved contractility, the most sensitive and relevant parameter, after 24 hours. Currently, the general consensus is that cardiac preparations rapidly deteriorate in culture and are only amenable to acute studies. No published method is able to maintain adult cardiac tissue outside the body for more than a few hours. Here we show for the first time the preservation of contractility together with a number of other parameters, including a full scale gene expression analysis and ultrastructure, in myocardial slices after 24 hours in culture.

In the new version, not only we show that we are able to maintain slices in culture for 24 hours, but we are also able to culture rabbit slices for up to 5 days without loss of contractility. We also include data using human failing myocardium, demonstrating that our method is amenable to translational studies. This is pioneering work, with the potential to change cardiac research and the approach to both *in vitro* and *in vivo* studies. Our platform is of interest to the entire cardiovascular research community, which is one of the largest and best funded, and utilisation of this platform is much needed and long overdue.

Yours sincerely,

Cesare Terracciano, MD, PhD

Reviewer 1 comments:

“This research consists of taking rat myocardial slices, placing them on a stretching scaffold in culture, keeping them alive and healthy for 24 hours, and measuring sarcomere length. It is still in the preliminary stage, with the authors not having tested their hypothesis beyond the 24hr stage. Without any additional data to substantiate these claims, it is not possible to determine the long term effects or usefulness of their research in human specimens. Further experiments and data are necessary.”

We thank the Reviewer for the insightful comment. The rat slice experiments were limited to 24 hours for practical reasons and dictated by the necessity to perform a very vast array of tests, in order to fully substantiate our claim that the tissue's structure and function are substantially preserved in the reported culture conditions before moving to longer periods of culture. We clearly demonstrate an optimal preload condition, which has not previously been described. The parameters measured are the most sensitive and relevant of heart function (direct force generation, contractile reserve, etc.). No other published paper, in our knowledge, reports such a detailed characterisation after 24 hours culture, hence our work is a very important breakthrough. Thanks to our findings, several experiments in cardiovascular research can now be confidently extended to a 24 hour time point rather than being limited to only few hours.

We agree with the Reviewer that longer time in culture would be desirable. For this reason, we have performed additional experiments and demonstrated that this technique can be used to maintain rabbit myocardial slices for a minimum of 5 days (Fig. 6B). We have demonstrated that both the maximal and basal contractility of rabbit myocardial slices are well preserved at 5 days.

In addition, given that myocardial slices are one of the very few methods that can be reliably used to study human myocardium ex vivo, we agree that demonstrating the use of this technique with human biopsies would be paramount. To address this comment, we have performed experiments using myocardial slices derived from explanted human heart failure specimens. Our data demonstrates optimal preservation of human myocardial slice contractility when cultured at $SL=2.2\mu m$ (same as in rat) (Fig. 6A).

Reviewer 2 comments:

“The manuscript describes an improved method to preserve structural and functional integrity of adult cardiac tissue slices (also called: organotypic culture).

The major finding is that a preload equivalent to a sarcomere length of $2.2\mu m$ is optimal for the maintenance of the tissue.

Several groups have tried to improve longevity of adult cardiac tissue slices before. While it is relatively easy to maintain embryonic or neonatal cardiac tissue viable for several days, adult cardiac tissue shows a rapid loss of viability and dedifferentiation, precluding chronic experiments. Therefore, experiments with adult cardiac tissue slices usually must be performed within hours or at least on the same day to be relevant.

In their manuscript, Watson et al. demonstrate structural and functional integrity for 24 hours. This is probably a further improvement and will ease acute experiments. However, it is still not enough to perform real chronic experiments. Several days would e.g. be necessary to analyse remodeling or to perform adenoviral-mediated gene transfer.

It is self-explanatory for an organ like the heart that “biomimetic electromechanical stimulation” is necessary for an advanced organotypic culture model of the heart and many other groups have already elaborated models to provide these physiological stimuli. For this reason, the major contribution of this manuscript is the implementation of sarcomere length

measurements and the thorough demonstration of the importance of this parameter to maintain the original differentiation of cardiac tissue using state of the art methods.

In contrast to many other groups that merely describe their model, we get a precise information on the optimal value for at least one parameter (sarcomere length). Though the result itself is not surprising, using it will help to optimise other groups' models as well and will be of interest to others in the community as well.

The authors thank the Reviewer for the detailed and perceptive review. The Reviewer supports the importance and relevance of this study and highlights the unmet need of a stable *in vitro* platform in cardiovascular research; a platform that we describe in our manuscript. We would like to refer the Reviewer to our replies to the Editors and Reviewer 1 to address some of his/her concerns.

In summary, every aspect of this study has been addressed before:

- Methodology of generating and maintaining viable cardiac tissue slices

The methodology of generating and maintaining viable cardiac tissue slices is not an aspect or goal of this study. We have reported this methodology already in our published paper by Watson SA et al., Nature Protocols 2017. The goal of the present paper is to determine whether myocardial slices (prepared as above) can be maintained in culture in a reliable way. This should permit relevant analysis beyond a few hours from preparation and away from a steeply declining performance, which is the norm in the majority of other cardiac preparations in culture. For the first time we provide very substantial evidence for this and our data are relevant because we show that we can extend culture to a minimum of 5 days in rabbit slices and employ the technique with human specimens.

- Importance of preload for the preservation of cardiac tissue

This comment contrasts with the Reviewer's remark where he/she states that no other study reports an accurate and quantitative validation of the importance of preload in the context of maintenance of cardiac tissue in culture. It has been known for a long time that preload is an important regulator of contractility (Frank-Starling Law) and that overload induces myocardial hypertrophy. Previous knowledge justifies and makes our hypothesis more solid, but this hypothesis has not been tested in slices or other complex cardiac models using physiological parameters. We therefore respectfully disagree that this aspect has been addressed before in the context of our hypothesis.

- Measurements of sarcomere length in vitro and in vivo

Again, this is not an aspect of this study but part of the methodology to quantify the amount of preload required during culture.

However, given the importance of this area of research, the number of papers published, combining this information and the accurate demonstration that we have reliable research tools for further studies is very important.

We thank the Reviewer for acknowledging that the data presented in our paper is very important.

Specific Comments for Author:

Title

My suggestion would be to talk about "tissue slices" instead of "issue" already in the title of the manuscript. In addition, the authors might think about focusing the title and the paper on "sarcomere length" and "preload".

- “cardiac tissue” has been replaced with “myocardial slices” in the title
- The authors do not feel that focusing on sarcomere length/preload is warranted, given that the other culture factors (electrical stimulation/oxygenation/etc.) are also vital for prolonged culture.

Introduction

Introduction – Line 78 & 109:

“With time, their morphology and function start to resemble those of developing cardiomyocytes found in the foetal and neonatal heart (14).”

“Chronic mechanical unloading of the heart induces a foetal phenotype, despite and unchanged humoral milieu, multicellularity, ECM and electrical stimulation (28).”

I would hesitate to compare dedifferentiated adult tissue slices with foetal or neonatal tissue slices. There are too many aspects that are different and primarily related to necrosis, apoptosis, fibroblast overgrowth, etc.

The authors use the following references to support their statements:

Reference 14:

Szibor M, Pöling J, Warnecke H, Kubin T, Braun T. Remodeling and dedifferentiation of adult cardiomyocytes during disease and regeneration. *Cell Mol Life Sci*. 2014;71(10):1907–16. This work is a review paper that also addresses dedifferentiation *in vitro* but not dedifferentiation of cardiac tissue slices and, although some aspects (e.g. gene expression) resemble a foetal phenotype, it doesn't support the author's simplification.

Reference 28:

Razeghi P, Sharma S, Ying J, Li Y-P, Stepkowski S, Reid MB, et al. Atrophic remodeling of the heart *in vivo* simultaneously activates pathways of protein synthesis and degradation. *Circulation* [Internet]. 2003 Nov 18 [cited 2018 Feb 2];108(20):2536–41. Available from: <http://www.ncbi.nlm.nih.gov/pubmed/14610007>

This manuscript deals with atrophic remodeling induced by heterotopic transplantation of the rat heart and is suited to highlight the importance of loading conditions but not the development of a foetal phenotype.

- We have removed the sentence “With time, their morphology and function start to resemble those of developing cardiomyocytes found in the foetal and neonatal heart (14).” and agree that while there are similarities between dedifferentiated adult and foetal cardiac tissue, there are also some differences.
- We have changed “induces a foetal phenotype” to “ induces atrophic remodelling” to more closely highlight the findings of Razeghi et al.

Introduction – Line 87:

“However, when pluripotent stem cell-derived cardiomyocytes are implanted into adult hearts they do undergo a phenotypic maturation (18).”

Reference 18:

Chong JJH, Yang X, Don CW, Minami E, Liu Y-W, Weyers JJ, et al. Human embryonic-stem-cell derived cardiomyocytes regenerate non-human primate hearts. *Nature* [Internet]. 2014 Jun 30 [cited 2018 Jan 7];510(7504):273–7. Available from: <http://www.nature.com/articles/nature13233>

This article provides evidence for a “progressive but incomplete maturation over a three month period” (cited from the abstract of the original article by Chong JJH et al.). The word ‘incomplete’ is important.

- We have included the phrase “progressive but incomplete” to describe the maturation of pluripotent stem cell-derived cardiomyocytes *in vivo*.

Results:

“Application of electromechanical stimulation to myocardial slices” I would have considered the first section of the “Results” (lines 127-161) including Figure 1 to be a part of the Methods section.

- The authors feel that the SL-% stretch relationship is an important result. Additionally, laser diffraction has never been demonstrated on myocardial slices before.

“Structure-Function Relationships: Contractility, Ca²⁺ handling & T-Tubular Structure”

(lines 162 ff. with Figure 2)

Line 164: “The maximum contractility (maximum force generation normalised to myocardial slice cross-sectional area).”

The term “contractility” is used by the authors for the force generated normalised to myocardial slice cross-sectional area. The term “contractility” should be replaced.

- The term “contractility” has been used extensively to describe the tension developed by myocardial fibers at a given preload and afterload. Both we and others have used this term to describe the maximum force generation normalised to myocardial slice cross-sectional area previously - Perbellini et al. *Cardiovascular Research* 2017, Brandenburger et al. *Cardiovascular Research* 2011.

Line 224 :

“T-tubule density, assessed by quantifying the density of caveolin-3 staining [...]” Ultrastructure, including T-tubular structure should have been assessed by electron microscopy.

- TEM has now been performed on 0Hr and 24Hr (all SLs) myocardial slices. The images show that the architecture of the myocardium and ultrastructure of cardiomyocytes are maintained in culture – Supplementary Figure 7
- We have assessed T-tubular structure using a methodology that has been previously used and described in several publications.
- The authors preferred to quantify structural properties with confocal microscopy as larger areas of each slice could be imaged and analysed compared to TEM.

“Conduction, Connexins, β -Adrenergic Stimulation & Arrhythmogenesis”

(lines 234 ff. with Figure 3)

Line 235: “The conduction velocity of cultured rat myocardial slices was assessed using a multielectrode array system as previously described (4). Reference (4): Camelliti P, Al-Saud SA, Smolenski RT, Al-Ayoubi S, Bussek A, Wettwer E, et al.

Even if the method is not described thoroughly, it is important to describe the parameter that was used for the MEA maps (time of maximum amplitude [i.e. U_{max}], time of maximum increase of voltage [i.e. $\Delta U/\Delta t$]).

- We have added a few words to the methods section to further describe the analysis: “A custom MATLAB (MathWorks) software was used to locate field potentials from recording electrodes, produce activation maps and calculate conduction velocities, using time of maximum amplitude, as previously described(3,4)”

Figure 3 A/B:

Information on the size of the square (cm) shown and the fiber orientation (e.g. an arrow) would be useful as well as using the same scaling for Figure 3A and 3B.

- Arrows have been added to Fig. 3A/B to show myocardial fiber direction. The size of the conduction map square (4.9x4.9cm) has been added to the figure legend.

Figure 3 E:

The images (pdf-file as well as WORD-file) provided to the reviewer do not provide enough resolution to judge the connexin 43 staining.

- The authors can provide the reviewer with higher quality images if required. Further analysis performed to assess connexin 43 heterogeneity and lateralisation can be found in supplementary figure 5.

“Cell Morphology, Mitochondria & Metabolism”

(lines 280 ff. with Figure 4)

Line 291: “It is likely these proliferating cells are cardiac fibroblasts [...]”

This is probably true but endothelial are probably also involved and could be easily stained e.g. with von Willebrand factor.

- As mentioned in the manuscript, we have previously published the same observation in dog and human HF myocardial slices. The experiments performed for that manuscript, including vWF stainings, indicated that these cells were primarily cardiac fibroblasts.
- Perbellini et al. 2017. Investigation of cardiac fibroblasts using myocardial slices. *Cardiovascular Research*. <https://doi.org/10.1093/cvr/cvx152>

“Electromechanical regulation of gene expression”

(lines 328 ff. with Figure 5)

Annotation:

The authors demonstrate a nice preservation during the first 24 hours. What would be important for future developments would have been an analysis of the changes thereafter. What are the changes that occurred despite preload and electrical stimulation after 24 hours in culture? The authors are asked to comment on this topic. It might even be important and helpful to provide some data on the unsolved problems.

- We have included 24hr data on Human HF myocardial slices, demonstrating optimal preservation of function at SL=2.2. Human slices are crucial for many translational approaches and this data is of interest to many groups.
- We have included data on our chronic culture studies of rabbit myocardial slices over 5 days. The data demonstrates that there is preservation of contractility at 5 days (no significant decline). We demonstrate that this is the case both in terms of basal and maximum contractility.
- The subtle decline that is observed may be attributed to the fact that the physiological environmental is not fully replicated. Load is static, not dynamic, and there is no humoral stimulation.
- We are not aware of any other approaches that can maintain any adult multicellular cardiac preparation for such a sustained period with minimal decline in function *in vitro*.

Discussion:

General comments:

The discussion is quite long, and many aspects are redundant. Therefore, it could be more concise.

- The entire discussion has been refined and shortened.

The most important information that is provided by the experiments is simply:

“we have demonstrated that the application of a preload equivalent to SL=2.2µm significantly reduces culture-induced remodelling compared to other preloads within the physiological range or unloaded. “(line 370)

The authors claim: “The ability to maintain myocardial slices in a steadier state *in vitro*, which

is not possible with current published techniques, provides a new and much needed platform for cardiovascular research.” (lines 374-375)

The authors try to elaborate that their model is new. Definitely, it is not a new but an improved model.

- We have removed the word “new” and replaced with “provides a significantly improved and much needed platform for cardiovascular research”

Therefore, it would be helpful to point out what they think are the most beneficial changes as compared to existing models. They focus on preloading but this, too, has been used before. Since many other scientists have used modifications of the model before, it would be helpful to discuss the changes that have been made that they think were most important.

- The second paragraph of the discussion now describes the specific benefits of our model:

The ability to maintain myocardial slices in a functional state *in vitro*, which is not possible with current published techniques, provides a significantly improved and much needed platform for cardiovascular research. The specific benefits of this improved model include prolonged preservation of the adult cardiac phenotype *in vitro*, intact native cardiac multicellularity and tissue architecture, acquisition of high-quality sensitive data on cardiac structure and function in real-time and an ability to investigate representative responses to pharmacological and mechanical loading stimuli *in vitro*. All current *in vitro* adult cardiac models undergo a rapid process of tissue dedifferentiation and loss of the adult cardiac properties (phenotype and function) in culture. Our group and others have previously shown that this also occurs when myocardial slices are cultured in unloaded conditions(7,9). For the first time, we have demonstrated that the loss of the adult cardiac phenotype can be significantly delayed in culture, with myocardial slices remaining highly functional for 5 days. Myocardial slices have preserved cardiac multicellularity and extracellular matrix, uniquely allowing the study of all cardiac cell populations in a representative environment. This is fundamental to a deeper understanding of cardiac physiology as the non-myocyte and non-cellular components of the myocardium play vital roles in the regulation of cardiomyocyte structure and function(20). In this study, we confirmed that electromechanical stimulation not only preserves cardiomyocyte function, but also maintains non-myocyte populations in a physiological state. We have confirmed that electromechanical stimulation prevents the massive proliferation of cardiac fibroblasts *in vitro* previously observed(9), while also maintaining a more physiological gene expression profile. The ultrathin nature of myocardial slices combines the benefits of a complex multicellular 3D preparation with the simplicity of acquiring structural and functional data from a 2D monolayer. In this study we have fully characterised both the structure and function of myocardial slices in culture using several high-sensitivity assays. We have measured direct force generation and contractile reserve, the most relevant parameters of cardiac function, and have demonstrated that both remain stable for 5 days. Additionally, we have demonstrated that cultured (24 hours) myocardial slices responded to isoproterenol stimulation in a comparable manner to gold-standard drug screening models(36,37), with both 0Hr and cultured myocardial slices (SL=1.8-2.4µm) displaying an EC₅₀ in the nanomolar ($\times 10^{-9}$ M) range. The model can also be used to investigate cardiac remodelling in response to both unloading and overload, which we have sensitively quantified using laser diffraction for the first time in myocardial slices.

Another very important point would have been to discuss the temporal limitations:

When experiments are done with this experimental set-up:

What is the time frame for physiologically relevant studies?

The authors may comment on this topic more precisely.

When we have the choice between different species.

Which species seems to be best suited to get relevant data?

Could the authors comment on the use of their method in mice.

- 4th paragraph of the discussion now addresses all of the points raised by the reviewer.

Although the discussion also mentions the limitations of the models, it would be helpful to

separate the limitations more clearly (e.g. by a headline, underlining).

- Limitations subtitle has been added

Methods

General comment:

Reproducibility of the model:

The authors describe an interesting further development of already existing tissue culture models and prove its relevance. For this reason, one of the most important topics is, that other researchers can reproduce the model. Therefore, a very precise and very detailed description of the model is mandatory. Although they might quote their previous work, other researchers should be able to use this technique with the help of this paper alone. Supplemental material might be helpful if the article would be too long otherwise.

- The authors would be happy to write a full protocol, which could exist as supplementary material, if the paper is accepted in this journal.
- The specific comments made by the reviewer have been addressed and are detailed below.

Specific comments:

Methods – Line 470

“Rat myocardial slices - Rats were sedated [...] “

Please specify the rats used preferably for adult slices (age, species, body weight).

- “Male Sprague Dawley rats, with a body mass between 300-500g”

Methods – Line 484

“The tissue block was sliced using a ceramic blade, longitudinal to myocardial fibre orientation and from endocardium down to the epicardium. The vibratome was set to vibrate at 80Hz, with an amplitude of 2mm and a Z-axis error <1.0µm.”

The authors should comment on the blades used:

Did they observe an improvement using the (very expensive) ceramic blades as compared to (less expensive) metal blades? The authors should specify the ceramic blade that was used.

- Ceramic blades were used to prepare myocardial slices in our published optimised protocol – Watson et al. 2017 *Nature Protocols*. Although being more expensive, ceramic blades can be reused many times (slicing almost every day, the average life of a ceramic blade is 3 months, which means that the actual price per experiment is almost negligible).

The fibre orientation changes from endocardium to epicardium. Was the orientation of the tissue block changed during slicing?

- Histological studies have demonstrated that myocardial fibers within the left ventricle are arranged into discrete sheets, which have an ordered laminar structure and are orientated transversally to the ventricular wall (LeGrice *et al.*, 1995). In our *Nature Protocols* paper, we described a method for dissecting the left ventricle of small and large mammalian hearts to ensure tissue flattening. When combined with mounting of the tissue block epicardium-down, sheets of myocardial fibers are aligned in the same plane of the vibratome blade, ensuring minimal tissue damage during slicing. This is particularly important when working with small mammalian hearts, which have a more pronounced ventricular curvature. Using this approach, damage incurred during slicing is strictly limited to the myocardial slice surface, with only ≈2-3% of the total cardiomyocyte population damaged during slicing.

Methods – Line 562

“Rat myocardial slices were field stimulated at 1Hz, 10ms, 10-30V and human HF & donor

slices were field stimulated at 0.5Hz, 10ms, 10-30V.”

The authors should comment on the voltage used.

Did they determine a threshold and stimulate with e.g. 150% or 200% of the threshold?

Did the slice respond in an all-or-nothing behaviour or was recruitment and force development influenced by the voltage used?

What kind of stimulation did they use?

The authors should specify the stimulator and the impulse precisely.

- The following sentences have been added to the Methods section: “Slices were electrically stimulated with a bipolar electrical stimulus. The voltage used was equivalent to 200% of the voltage required to capture the slice and as voltage was increased recruitment was observed”.

Methods – Line 566

A video of the contraction of human HF 566 and rat myocardial slices can be found here:

<https://images.nature.com/original/natureassets/nprot/journal/v12/n12/extref/nprot.2017.139-sv8.mp4>

This video is helpful for all researchers that would like to use this model. The video would be even better if anode and cathode would be labelled.

- The video is part of a previous publication and cannot be edited.

Methods – Line 612

Slices were stimulated longitudinal and transverse to known myofibril direction. A custom MATLAB (MathWorks) software was used to locate field potentials from recording electrodes, produce activation maps and calculate conduction velocities as previously described(3,4).

Important issue:

Since the authors submit this manuscript to a high-ranked journal, I would have expected ultrastructural images as well. I would have considered this necessary to demonstrate structural integrity.

- Representative TEM images of 0Hr slices and slices cultured for 24hrs with electromechanical stimulation have now been included – please see supplementary figure 7.

Minor Comments:

Abstract – Line 38:

The abbreviation „SL“ („sarcomere length“) should be defined in the abstract as well.

- Added

The authors should use one spelling for the word „caliper“ respectively „calliper“.

- Changed to “caliper”

Figure 1: The authors use „gray“ in the figure and „grey“ in the figure legend.

- Changed to “gray”

Reviewers' comments:

Reviewer #1 (Remarks to the Author):

The revision is unresponsive to my concerns. The experiments are very short-term, and no new data are presented on human slices.

Reviewer #2 (Remarks to the Author):

Comments to the authors:

Title/Authors:

The number of corresponding authors was decreased from three to two authors. Could the authors comment on the reason, please.

Abstract:

Methods & Results:

The authors changes:

"We utilised rat, rabbit and human heart failure myocardial slices [...]" instead of

"We used myocardial slices [...]"

The abstract is not really helpful to summarize the work of the authors. It is not clear, what tissue they used for which experiment.

This is misleading and gets even more confusing / misleading by the sentence that was added at the end of the Methods & Results section:

"The platform was used to maintain rabbit myocardial slices contracting in a steady-state for 5 days."

The vast majority refers to a time span of 24 hours, adding this sentence is confusing / misleading.

The abstract definitely needs another revision!

Introduction:

Line 146

Probably a typing error: "dedifferentiation" should replace "differentiation"

Results:

"Electromechanical stimulation of human myocardial slices & chronic culture"

Since several species are used, it would be far easier for the reader if the species would be clarified in the headline:

"Electromechanical stimulation of human myocardial slices & chronic culture of rabbit myocardial slices"

Figure 3 6B:

If the maximum contractility was measured in the same slices, it would make more sense to show the data of n=6 slices

instead of "increasing" the number of observations without corresponding observations at the other two time points.

This should be changed.

Discussion:

Line 485: "myocardial slices remaining highly functional for 5 days"

Again, it should be clarified that the authors talk about rabbit tissue, only.

Conclusion:

Line 565: "Breakthrough" ?

Response to Reviewers:

Reviewer #1 (Remarks to the Author):

The revision is unresponsive to my concerns. The experiments are very short-term, and no new data are presented on human slices.

The authors were disheartened that Reviewer 1 failed to acknowledge the addition of new data collected using human heart failure specimens and 5 day rabbit myocardial slice experiments. The authors are concerned that an objective and fair review process failed to take place.

Reviewer #2 (Remarks to the Author):

Comments to the authors:

Title/Authors:

The number of corresponding authors was decreased from three to two authors. Could the authors comment on the reason, please.

The first author of the manuscript has now finished their PhD and has left the Terracciano laboratory. They agree that they no longer need to be a corresponding author.

Abstract:

Methods & Results:

The authors changes:

"We utilised rat, rabbit and human heart failure myocardial slices [...]" instead of
"We used myocardial slices [...]"

The abstract is not really helpful to summarize the work of the authors. It is not clear, what tissue they used for which experiment.

This is misleading and gets even more confusing / misleading by the sentence that was added at the end of the Methods & Results section:

"The platform was used to maintain rabbit myocardial slices contracting in a steady-state for 5 days."

The vast majority refers to a time span of 24 hours, adding this sentence is confusing / misleading.

The abstract definitely needs another revision!

The abstract has been completely revised to ensure clarity surrounding tissue-type and time points. The authors hope the reviewer finds the abstract more useful.

Introduction:

Line 146

Probably a typing error: "dedifferentiation" should replace "differentiation"

This error has been changed.

Results:

"Electromechanical stimulation of human myocardial slices & chronic culture"

Since several species are used, it would be far easier for the reader if the species would be clarified in the headline:

"Electromechanical stimulation of human myocardial slices & chronic culture of rabbit myocardial slices"

This has been changed and the species used for each experiment has been highlighted throughout the manuscript.

Figure 3 6B:

If the maximum contractility was measured in the same slices, it would make more sense to show the data of n=6 slices

instead of "increasing" the number of observations without corresponding observations at the other two time points.

This should be changed.

The data at each time point was collected from different myocardial slices. Maximum contractility testing results in the destruction of the specimen. The basal contractility data is derived from the 5 day experiments, which has been highlighted in the figure legend.

Discussion:

Line 485: "myocardial slices remaining highly functional for 5 days"

Again, it should be clarified that the authors talk about rabbit tissue, only.

This has been clarified in the text.

Conclusion:

Line 565: "Breakthrough" ?

"Breakthrough" has been replaced with "significant advance"